# What Do GNNs Actually Learn? Towards Understanding their Representations

## Abstract

Although prior work has shed light on the expressiveness of Graph Neural Networks (GNNs) (i. e., whether they can distinguish pairs of non-isomorphic graphs), it is still not clear what structural information is encoded into the node representations that are learned by those models. In this paper, we address this gap by studying the node representations learned by four standard GNN models. We find that some models produce identical representations for all nodes, while the representations learned by other models are linked to some notion of walks of specific length that start from the nodes. We establish Lipschitz bounds for these models with respect to the number of (normalized) walks. Additionally, we investigate the influence of node features on the learned representations. We find that the representations learned at the $k$-the layer of the models are related to the initial features of nodes that can be reached in exactly $k$ steps. We bound the Lipschitz constant of these models with respect to an optimization problem matching nodes' sets of walks. Our theoretical analysis is validated through experiments on synthetic and real-world datasets. We also apply our findings to understand the phenomenon of oversquashing that occurs in GNNs.

## 1 Introduction

Graphs arise naturally in a wide variety of domains such as in bio- and chemo-informatics (Stokes et al., 2020), in social network analysis (Easley & Kleinberg, 2010) and in information sciences (Hogan et al., 2021). There is thus a need for machine learning algorithms that can operate on graph-structured data, i. e., algorithms that can exploit both the information encoded in the graph structure but also the information contained in the node and edge features. Recently, graph neural networks (GNNs) emerged as a very promising method for learning on graphs, and have driven the rapid progress in the field of graph representation learning (Wu et al., 2020).

Even though different types of GNNs were proposed in the past years, message passing models undoubtedly seem like a natural approach to the problem. These models, known as message passing neural networks (MPNNs) (Gilmer et al., 2017) employ a message passing (or neighborhood aggregation) procedure where each node aggregates the representations of its neighbors along with its own representation to produce new updated representations. For graph-related tasks, MPNNs usually apply some permutation invariant readout function to the node representations to produce a representation for the entire graph. The family of MPNNs has been studied a lot in the past few years, and there are now available dozens of instances of this family of models. A lot of work has focused on investigating the expressive power of those models. It was recently shown that standard MPNNs are at most as powerful as the Weisfeiler-Leman algorithm in terms of distinguishing non-isomorphic graphs (Xu et al., 2019; Morris et al., 2019).

The recent success of GNNs put graph kernels, another approach for graph–based machine learning, into the shade. Unlike GNNs, graph kernels generate representations (implicit or explicit) that consist of substructures of graphs. Such substructures include random walks (Kashima et al., 2003; Gärtner et al., 2003), shortest paths (Borgwardt & Kriegel, 2005) and subgraphs (Shervashidze et al., 2009; Kriege & Mutzel, 2012). Therefore, the properties and the graph representations produced by graph kernels are fully-understood. This is not however the case for MPNNs since, despite the great activity in the field, still little is known about the properties of graphs that are captured in the representations learned by those models.

In this paper, we fill this gap by studying the node representations learned by MPNNs. We first investigate what structural properties of graphs are captured in the learned representations of standard models. To that end, we annotate all nodes with the same features, and show that GAT (Veličković et al., 2018) and DGCNN (Zhang et al., 2018) embed all nodes into the same vector, thus they capture no structural properties of the neighborhoods of nodes. Furthermore, we show that the representations that emerge at the $k$-th layer of GCN (Kipf & Welling, 2017) and GIN (Xu et al., 2019) are related to some notion of walks of length $k$ over the input graph. We bound the Lipschitz constant of those models with respect to the sum of (normalized) walks. This suggests that MPNNs suffer from the following limitation: structurally dissimilar nodes can have similar representations at some layer $k$ where $k > 1$. We also study the impact of node features on the learned representations. We show that the node representations at the $k$-th layer of GCN, DGCNN, GAT and GIN are all related to the initial features of the nodes that can be reached in exactly $k$ steps from the node. We bound the Lipschitz constant of those models with respect to the solution of some optimization problem that given two nodes matches walks of the one node with walks of the other. We verify our theoretical analysis in experiments conducted on synthetic and real-world datasets. We also study the problem of oversquashing (Alon & Yahav, 2021) from the lens of our theoretical findings.

## 2 RELATED WORK

While GNNs have been around for decades (Sperduti & Starita, 1997; Scarselli et al., 2009; Micheli, 2009), it is only in recent years that the scientific community became aware of their power and potential. The increased scientific activity in the field led to the development of a large number of models (Bruna et al., 2014; Li et al., 2015; Duvenaud et al., 2015; Atwood & Towsley, 2016; Defferrard et al., 2016). Those models were categorized into spectral and spatial approaches depending on which domain the convolutions (neighborhood aggregations) were performed. Later, it was shown that all these models follow the same design principle and can be seen as instances of a single common framework (Gilmer et al., 2017). These models, known as message passing neural networks (MPNNs), use a message passing scheme where nodes iteratively aggregate feature information from their neighbors. Then, to compute a representation for the entire graph, MPNNs typically employ some permutation invariant readout function which aggregates the representations of all the nodes of the graph. The family of MPNNs has been studied a lot in the past few years and there have been proposed several extensions and improvements to the MPNN framework. Most studies have focused on the message passing procedure and have proposed more expressive or permutation sensitive aggregation functions (Murphy et al., 2019; Seo et al., 2019; Chatzianastasis et al., 2022; Buterez et al., 2022), schemes that incorporate different local structures or high-order neighborhoods (Jin et al., 2020; Abu-El-Haija et al., 2019), non-Euclidean geometry approaches (Chami et al., 2019), while others have focused on efficiency (Gallicchio & Micheli, 2020). Fewer works have focused on the pooling phase and have proposed more advanced strategies for learning hierarchical graph representations (Ying et al., 2018; Gao & Ji, 2019). Note also that not all GNNs belong to the family of MPNNs (Niepert et al., 2016; Nikolentzos & Vazirgiannis, 2020; Nikolentzos et al., 2023b)

A considerable amount of recent work has focused on characterizing the expressive power of GNNs. Most of these studies compare GNNs against the WL algorithm and its variants (Kiefer, 2020) to investigate what classes of non-isomorphic graphs they can distinguish. For instance, it has been shown that standard GNNs are not more powerful than the 1-WL algorithm (Xu et al., 2019; Morris et al., 2019). Other studies capitalized on high-order variants of the WL algorithm to derive new models that are more powerful than standard MPNNs (Morris et al., 2019; 2020). Recent research has investigated the expressive power of $k$-order GNNs in terms of their ability to distinguish non-isomorphic graphs. In particular, it has been shown that $k$-order GNNs are at least as powerful as the $k$-WL test in this regard (Maron et al., 2019). Recently, various approaches have been proposed to enhance the expressive power of GNNs beyond that of the WL test. These include encoding vertex identifiers (Vignac et al., 2020), incorporating all possible node permutations (Murphy et al., 2019; Dasoulas et al., 2020), using random features (Sato et al., 2021; Abboud et al., 2020), utilizing node features (You et al., 2021), incorporating spectral information (Balcilar et al., 2021), utilizing simplicial and cellular complexes (Bodnar et al., 2021b;a) and directional information (Beaini et al., 2021). It has also been shown that extracting and processing subgraphs can further enhance the expressive power of GNNs (Nikolentzos et al., 2020; Zhang & Li, 2021; Bevilacqua et al., 2021) . For instance, it has been suggested that expressive power of GNNs can be increased by aggregating

the representations of subgraphs produced by standard GNNs, which arise from removing one or more vertices from a given graph (Cotta et al., 2021; Papp et al., 2021). The above studies mainly focus on whether GNNs can distinguish pairs of non-isomorphic graph. However, it still remains unclear what kind of structural information is encoded into the node representations learned by GNNs. Some recent works have proposed models that aim to learn representations that preserve some notion of distance of nodes (Nikolentzos et al., 2023a), however, they do not shed light into the representations generated by standard models. The work closest to ours is the one proposed by Chuang & Jegelka (2022), where the authors propose the Tree Mover's Distance, a pseudometric for node-attributed graphs, and study its relation to the generalization of GNNs. Our work is also related to the work of Xu et al. (2018) where the authors use the concept of walks to define the effective range of nodes that any given node's representation draws from. However, while this work studies the range of nodes whose features affect a given node's representation, we focus on the exact node representations that are learned by the model and the distances between them. Finally, Yehudai et al. (2021) capitalize on local computation trees and graph patterns similar to the ones studied in this paper to investigate the GNNs' ability to generalize to larger graphs

## 3 PRELIMINARIES

### 3.1 NOTATION

Let $\mathbb{N}$ denote the set of natural numbers, i.e., $\{1, 2, \ldots\}$. Then, $[n] = \{1, \ldots, n\} \subset \mathbb{N}$ for $n \geq 1$. Let also $\{\!\{\}\!\}$ denote a multiset, i.e., a generalized concept of a set that allows multiple instances for its elements. Let $G = (V, E)$ be an undirected graph, where $V$ is the vertex set and $E$ is the edge set. We will denote by $n$ the number of vertices and by $m$ the number of edges, i.e., $n = |V|$ and $m = |E|$. The adjacency matrix $\mathbf{A} \in \mathbb{R}^{n \times n}$ is a symmetric matrix used to encode edge information in a graph. The elements of the $i^{\text{th}}$ row and $j^{\text{th}}$ column is equal to 1 if there is an edge between $v_i$ and $v_j$, and 0 otherwise. Let $\mathcal{N}(v)$ denote the the neighbourhood of vertex $v$, i.e., the set $\{u \mid \{v, u\} \in E\}$. The degree of a vertex $v$ is $d(v) = |\mathcal{N}(v)|$. A walk is a sequence of nodes $(v_1, v_2, \ldots, v_{k+1})$ where $v_i \in V$ and $(v_i, v_{i+1}) \in E$ for $1 \leq i \leq k$. The length of the walk is equal to the number of edges in the sequence, i.e., $k$ in the above case. We denote by $w_v^{(k)}$ the number of walks of length $k$ starting from node $v$. Let $\tilde{w}_v^{(k)}$ denote the sum of normalized walks of length $k$ where each walk $(v_1, v_2, \ldots, v_k)$ is normalized as follows $1/\big((1+d(v_2))\ldots(1+d(v_{k-1}))\sqrt{(1+d(v_1))(1+d(v_k))}\big)$. Given a walk $\mathsf{w} = (v_1, v_2, \ldots, v_{k+1})$ of length $k$, we denote by $\bar{w}_{\mathsf{w}}^{(k)}$ the probability of getting from node $v_1$ to node $v_{k+1}$ via walk $\mathsf{w}$. Finally, let $\tilde{w}_{\mathsf{w}}^{(k)}$ denote a normalization term for walk $\mathsf{w}$ computed as follows $\tilde{w}_{\mathsf{w}}^{(k)} = 1/\big((1+d(v_2))\ldots(1+d(v_k))\sqrt{(1+d(v_1))(1+d(v_{k+1}))}\big)$.

### 3.2 MESSAGE PASSING NEURAL NETWORKS

As already discussed, most GNNs can be unified under the framework MPNN framework (Gilmer et al., 2017). These models follow a neighborhood aggregation scheme, where each node representation is updated based on the aggregation of its neighbors representations. Let $\mathbf{h}_v^{(0)}$ denote node $v$'s initial feature vector. Then, for a number $K$ of iterations, MPNNs update node representations as follows:

$$\mathbf{m}_v^{(k)} = \text{AGGREGATE}^{(k)}\left(\{\!\{\mathbf{h}_u^{(k-1)} | u \in \mathcal{N}(v)\}\!\}\right), \quad \mathbf{h}_v^{(k)} = \text{COMBINE}^{(k)}\left(\mathbf{h}_v^{(k-1)}, \mathbf{m}_v^{(k)}\right)$$

where AGGREGATE$^{(k)}$ is a permutation invariant function. By defining different AGGREGATE$^{(k)}$ and COMBINE$^{(k)}$ functions, we obtain different MPNN instances. In this study, we consider the neighborhood aggregation schemes of four models, namely (1) Graph Convolution Network (GCN) (Kipf & Welling, 2017); (2) Deep Graph Convolutional Neural Network (DGCNN) (Zhang et al., 2018); (3) Graph Attention Network (GAT) (Veličković et al., 2018); and (4) Graph Isomorphism Network (GIN) (Xu et al., 2019). The aggregation schemes of the four models are illustrated in Table 1.

For node-level tasks, final nodes representation $\mathbf{h}_v^{(K)}$ can be directly passed to a fully-connected layer for prediction. For graph-level tasks, a graph representation is obtained by aggregating its

Table 1: Neighborhood aggregation schemes of the four considered models.

| Model | Update Equation |
|---|---|
| **GCN** | $\mathbf{h}_v^{(k)} = \text{ReLU}\left( \sum_{u \in \mathcal{N}(v) \cup \{v\}} \frac{\mathbf{W}^{(k)} \mathbf{h}_u^{(k-1)}}{\sqrt{(1+d(v))(1+d(u))}} \right)$ |
| **DGCNN** | $\mathbf{h}_v^{(k)} = f\left( \sum_{u \in \mathcal{N}(v) \cup \{v\}} \frac{1}{d(v)+1} \mathbf{W}^{(k)} \mathbf{h}_u^{(k-1)} \right)$ |
| **GAT** | $\mathbf{h}_v^{(k)} = \sigma\left( \sum_{u \in \mathcal{N}(v)} \alpha_{vu} \mathbf{W}^{(k)} \mathbf{h}_u^{(k-1)} \right)$ |
| **GIN-$\epsilon$** | $\mathbf{h}_v^{(k)} = \text{MLP}^{(k)}\left( \left(1 + \epsilon^{(k)}\right) \mathbf{h}_v^{(k-1)} + \sum_{u \in \mathcal{N}(v)} \mathbf{h}_u^{(k-1)} \right)$ |

nodes final representations: $\mathbf{h}_G = \text{READOUT}\left( \{\!\{ \mathbf{h}_v^{(K)} | v \in G \}\!\} \right)$. The READOUT function is typically a differentiable permutation invariant function such as the sum or mean operator.

## 4 WHAT DO MPNNs ACTUALLY LEARN?

We next study the node representations learned by the considered model. We focus on two different scenarios: (1) when nodes are not annotated with any features; and (2) when nodes are annotated with discrete node labels or continuous attributes. The first scenario will shed light on what kind of structural properties of graphs are captured in the representations learned by GNNs. In the second part, we will investigate what is the impact of the initial node features on the learned representations.

### 4.1 CAN MPNNs CAPTURE THE STRUCTURE OF GRAPHS?

We next investigate what structural properties of nodes these four considered models can capture. Nodes are usually annotated with features that reveal information about their neighborhoods. Such features include their degree or even more sophisticated features such as counts of certain substructures (Bouritsas et al., 2022) or those extracted from Laplacian eigenvectors (Dwivedi et al., 2020). We are interested in identifying properties that are captured purely by these models. Thus, we assume that no such features are available, and we annotate all nodes with the same feature vector or scalar.

**Theorem 4.1.** *Let $\mathcal{G} = \{G_1, \ldots, G_N\}$ be a collection of graphs. Let also $\mathcal{V} = V_1 \cup \ldots \cup V_N$ denote the set that contains the nodes of all graphs. All nodes are initially annotated with the same representation. Without loss of generality, we assume that they are annotated with a single feature equal to 1, i.e., $\mathbf{h}_v^{(0)} = 1 \; \forall v \in \mathcal{V}$. Then, after $k$ neighborhood aggregation layers:*

1. *DGCNN and GAT both map all nodes to the same representation, i.e., $\mathbf{h}_v^{(k)} = \mathbf{h}_u^{(k)} \; \forall v, u \in \mathcal{V}$.*

2. *GCN maps nodes to representations related to the sum of normalized walks of length $k$ starting from them:*

$$\left\| \mathbf{h}_v^{(k)} - \mathbf{h}_u^{(k)} \right\|_2 \leq \prod_{i=1}^{k} L_f^{(i)} \left\| \tilde{w}_v^{(k)} - \tilde{w}_u^{(k)} \right\|_2$$

*where $L_f^{(i)}$ denotes the Lipschitz constant of the fully-connected layer of the $i$-th neighborhood aggregation layer.*

3. *Under mild assumptions (biases of MLPs are ignored), GIN-0 maps nodes to representations that capture the number of walks of length $k$ starting from them:*

$$\left\| \mathbf{h}_v^{(k)} - \mathbf{h}_u^{(k)} \right\|_2 \leq \prod_{i=1}^{k} L_f^{(i)} \left\| w_v^{(k)} - w_u^{(k)} \right\|_2$$

*where $L_f^{(i)}$ denotes the Lipschitz constant of the MLP of the $i$-th neighborhood aggregation layer.*

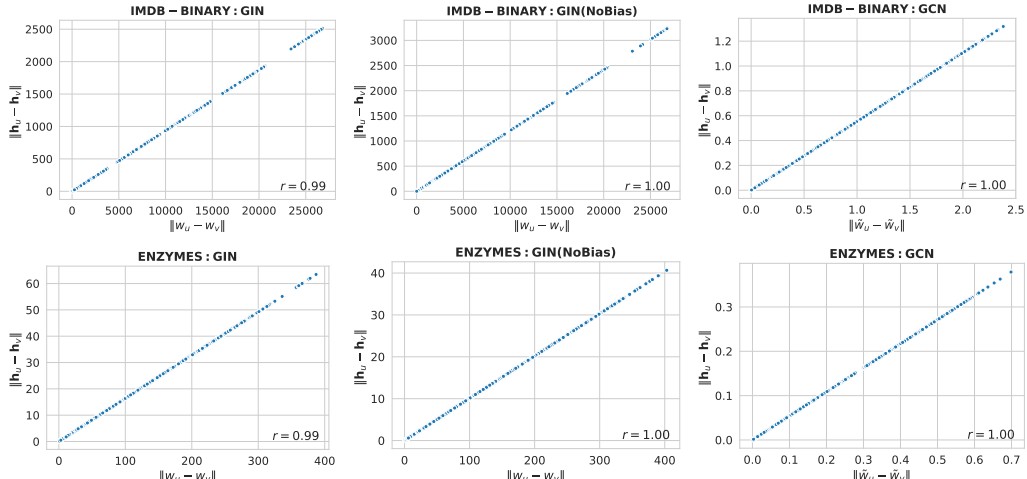

Figure 1: Euclidean distances of the representations generated at the third layer of the different models vs. Euclidean distances of the number of walks (or sum of normalized walks) of length 3 starting from the different nodes.

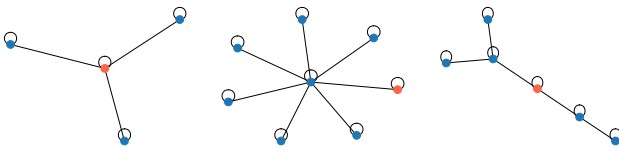

Figure 2: The number of walks of length 2 starting from the red nodes of the three graphs is equal to 10. These three nodes could be embedded closely to each other even though they are structurally dissimilar.

The above result highlights the limitations of the considered models. Specifically, our results imply that the DGCNN and GAT models encode no structural information of the graph into the learned node representations. Furthermore, combined with a sum readout function, these representations give rise to a graph representation that can only count the number of nodes of the graph. If the readout function is the mean operator, then all graphs are embedded into the same vector. With regards to the other two models, we have bounded the Lipschitz constant of GIN-0 and GCN with respect to the number of walks and sum of normalized walks starting from the different nodes, respectively.

To experimentally verify the above theoretical results, we trained the GIN-0 and GCN models on the IMDB-BINARY and the ENZYMES graph classification datasets. For all pairs of nodes, we computed the Euclidean distance of the number of walks (resp. sum of normalized walks) of length 3 starting from them. We also computed the Euclidean distance of the representations of the nodes that emerge at the corresponding (i.e., third) layer of GIN-0 (resp. GCN). We finally computed the correlation of the two collections of Euclidean distances and the results are given in Figure 1. Clearly, the results verify our theoretical results. The distance of the number of walks is perfectly correlated with the distance of the representations generated by GIN-0 with no biases, while the distance of the sum of normalized walks is perfectly correlated with the distance of the representations produced by GCN. We also computed the Euclidean distance of the representations of the nodes that emerge at the third layer of the standard GIN-0 model (with biases), and we compared them against the distances of the number of walks. We can see that on both datasets, the emerging correlations are very high (equal to $0.99$). We observed similar values of correlation on other datasets as well, which indicates that the magnitude of the bias terms of the MLPs might be very small and that our assumption of ignoring biases is by no means unrealistic.

Based on the above theoretical and empirical findings, it is clear that two nodes can have dissimilar representations at the $k$-th layer, but obtain similar representations at the $(k+1)$-th layer of some

MPNN model. For instance, for GIN-0, this can be the case if the two nodes have different numbers of walks of length $k$, but similar numbers of walks of length $k + 1$. We give in Figure 2 an example of three nodes (the three red nodes) that have structurally dissimilar neighborhoods, but their representations produced by GIN-0 after two neighborhood aggregation layers are very similar to each other (or identical in the case where biases are omitted). In all three cases, the number of walks of length 2 starting from the red nodes is equal to 10. It is also clear that the three nodes have different values of degree from each other.

## 4.2 WHAT IS THE ROLE OF THE INITIAL NODE FEATURES?

We next investigate what is the impact of the initial node features on the learned node representations. We assume that the nodes of all graphs are annotated with $d$-dimensional feature vectors. These feature vectors usually do not capture structural properties of nodes. Instead, they provide additional information about nodes which needs to be taken into account by the model. For instance, in chemo-informatics, such features could correspond to continuous atomic properties, while in the case of social networks, they could correspond to vector representations of text.

We next introduce some notations and preliminary concepts which will be used later. More specifically, we first define for each node $v$ of a graph some sets that contain the weighted representations of the nodes that can be reached in exactly $k$ steps. Let $\mathcal{W}_v^{(k)}$ denote a set of ordered pairs where each pair contains a walk of length $k$ starting from node $v$ along with the initial node feature of the last node of the walk. Formally, $\mathcal{W}_v^{(k)} = \left\{ (\mathsf{w}_1, \mathbf{h}_{\mathsf{w}_1}), \ldots, (\mathsf{w}_m, \mathbf{h}_{\mathsf{w}_m}) \right\}$ where $m$ is the number of walks of length $k$ that start from node $v$. Given a walk $\mathsf{w}$, $\mathbf{h}_{\mathsf{w}}$ denotes the initial node feature of the last node of the walk. For example, for some walk $\mathsf{w} = (v_1, v_2, \ldots, v_{k+1})$, we have that $\mathbf{h}_{\mathsf{w}} = \mathbf{h}_{v_{k+1}}^{(0)}$. We also define two other sets $\bar{\mathcal{W}}_v^{(k)}$ and $\tilde{\mathcal{W}}_v^{(k)}$ which also contain all walks of length $k$ starting from node $v$ along with the normalized representation of the last visited node. Thus, we have that $\bar{\mathcal{W}}_v^{(k)} = \left\{ (\mathsf{w}_1, \bar{\mathbf{h}}_{\mathsf{w}_1}), \ldots, (\mathsf{w}_m, \bar{\mathbf{h}}_{\mathsf{w}_m}) \right\}$ where for some walk $\mathsf{w} = (v_1, v_2, \ldots, v_{k+1})$, $\bar{\mathbf{h}}_{\mathsf{w}} = \bar{w}_{\mathsf{w}}^{(k)} \mathbf{h}_{v_{k+1}}^{(0)}$ and as discussed in section 3, $\bar{w}_{\mathsf{w}}^{(k)}$ is equal to the probability of getting from node $v_1$ to node $v_{k+1}$ via walk $\mathsf{w}$. We also have that $\tilde{\mathcal{W}}_v^{(k)} = \left\{ (\mathsf{w}_1, \tilde{\mathbf{h}}_{\mathsf{w}_1}), \ldots, (\mathsf{w}_m, \tilde{\mathbf{h}}_{\mathsf{w}_m}) \right\}$ where for some walk $\mathsf{w} = (v_1, v_2, \ldots, v_{k+1})$, $\tilde{\mathbf{h}}_{\mathsf{w}} = \tilde{w}_{\mathsf{w}}^{(k)} \mathbf{h}_{v_{k+1}}^{(0)}$ where $\tilde{w}_{\mathsf{w}}^{(k)}$ is computed as follows $\tilde{w}_{\mathsf{w}}^{(k)} = 1 / \left( (1+d(v_2))\ldots(1+d(v_k)) \sqrt{(1+d(v_1))(1+d(v_{k+1}))} \right)$. Given two walks $\mathsf{w}_1$ and $\mathsf{w}_2$, let $\mathrm{CSL}(\mathsf{w}_1, \mathsf{w}_2)$ denote the length of the common subsequence of nodes of the two walks starting from the first node. For instance, let $\mathsf{w}_1 = (v_1, v_2, v_4, v_8, \ldots, v_k)$ and $\mathsf{w}_2 = (v_1, v_2, v_4, v_{10}, \ldots, v_k)$. Then, $\mathrm{CSL}(\mathsf{w}_1, \mathsf{w}_2) = 3$. Let also $\mathsf{w}_3 = (v_1, v_2, v_4, v_5, \ldots, v_k)$ and $\mathsf{w}_4 = (v_1, v_3, v_4, v_5, \ldots, v_k)$. Then, $\mathrm{CSL}(\mathsf{w}_3, \mathsf{w}_4) = 1$.

We next introduce the walk distance (WD), a distance for sets of walks that corresponds to the solution of an optimization problem. Specifically, given two sets of walks $\mathcal{W}_v^{(k)}$ and $\mathcal{W}_{v'}^{(k)}$ from nodes $v$ and $v'$, respectively, where $|\mathcal{W}_v^{(k)}| = n$ and $|\mathcal{W}_{v'}^{(k)}| = m$, we will denote by $\mathrm{WD}(\mathcal{W}_v^{(k)}, \mathcal{W}_{v'}^{(k)})$ the solution of the following problem:

$$\mathrm{WD}(\mathcal{W}_v^{(k)}, \mathcal{W}_{v'}^{(k)}) = \min_{\mathbf{T}} \left( \sum_{i=1}^{n} \sum_{j=1}^{m} \mathbf{T}_{ij} ||\mathbf{h}_{\mathsf{w}_i} - \mathbf{h}_{\mathsf{w}_j'}||_2 + \sum_{j=1}^{m} \mathbf{T}_{(n+1)j} ||\mathbf{h}_{\mathsf{w}_j'}||_2 + \sum_{i=1}^{n} \mathbf{T}_{i(m+1)} ||\mathbf{h}_{\mathsf{w}_i}||_2 \right)$$

(1)

$$\text{s.t.} \quad \mathbf{T} \in \{0, 1\}^{(n+1) \times (m+1)}, \quad \sum_{j}^{m+1} \mathbf{T}_{ij} = 1, \forall i \in [n], \quad \sum_{i}^{n+1} \mathbf{T}_{ij} = 1, \forall j \in [m],$$

$$\mathbf{T}_{ij} + \mathbf{T}_{\imath\jmath} \leq 1, \forall i, \imath \in [n] \text{ and } \forall j, \jmath \in [m] \text{ where } \mathrm{CSL}(\mathsf{w}_i, \mathsf{w}_\imath) \neq \mathrm{CSL}(\mathsf{w}_j', \mathsf{w}_\jmath')$$

Note that we can use the same formulation and compute the distance of the other sets of walks that were defined above (i. e., $\mathrm{WD}(\bar{\mathcal{W}}_v^{(k)}, \bar{\mathcal{W}}_{v'}^{(k)})$ and $\mathrm{WD}(\tilde{\mathcal{W}}_v^{(k)}, \tilde{\mathcal{W}}_{v'}^{(k)})$) The last constraint ensures that each non-terminal node of the walks of the one set is matched with at most a single non-terminal node of the walks of the other set. To make this clear, we provide an example in Figure 4. Suppose that $\mathbf{h}_{v_6}^{(0)} = \mathbf{h}_{u_7}^{(0)}$, $\mathbf{h}_{v_7}^{(0)} = \mathbf{h}_{u_8}^{(0)}$ and $\mathbf{h}_{v_8}^{(0)} = \mathbf{h}_{u_9}^{(0)}$. Therefore, we have that $\mathbf{h}_{\mathsf{w}_1} = \mathbf{h}_{\mathsf{w}_1'}$, $\mathbf{h}_{\mathsf{w}_2} = \mathbf{h}_{\mathsf{w}_2'}$

and $\mathbf{h}_{w_3} = \mathbf{h}_{w'_3}$. Then, we can match walk $w_3$ with walk $w'_3$ (i.e., $\mathbf{T}_{33} = 1$). Suppose we also match walk $w_1$ with walk $w'_1$ (i.e., $\mathbf{T}_{11} = 1$). Then, we cannot match walk $w_2$ with walk $w'_2$ since $\mathrm{CSL}(w_1, w_2) = 3$ while $\mathrm{CSL}(w'_1, w'_2) = 2$, and thus $\mathbf{T}_{11} + \mathbf{T}_{22} \leq 1$.

Although one could solve the above optimization problem to compute the distance of two nodes, its computational complexity is prohibitive for real-world problems. Determining the exact complexity of the above problem is out of the scope of this paper. We only need to note that the number of walks increases significantly as $k$ increases in the case of dense graphs. For instance, in a complete graph, exhaustive enumeration of the walks that start from some node is prohibitively expensive, scaling as $\mathcal{O}(n^k)$ where $n$ is the number of nodes of the graph.

We now provide the main result of this subsection.

**Theorem 4.2.** *Let $\mathcal{G} = \{G_1, \ldots, G_N\}$ be a collection of graphs. Let also $\mathcal{V} = V_1 \cup \ldots \cup V_N$ denote the set that contains the nodes of all graphs. We assume that the nodes of all $N$ graphs are annotated with $d$-dimensional feature vectors, i.e., $\mathbf{h}_v^{(0)} \mathbb{R}^d \ \forall v \in \mathcal{V}$. Then, after $k$ neighborhood aggregation layers:*

1. *DGCNN and GAT map nodes to representations such that:*

$$\left\| \mathbf{h}_v^{(k)} - \mathbf{h}_u^{(k)} \right\|_2 \leq \prod_{i=1}^{k} L_f^{(i)} WD(\bar{\mathcal{W}}_v^{(k)}, \bar{\mathcal{W}}_u^{(k)})$$

   *where $L_f^{(i)}$ denotes the Lipschitz constant of the fully-connected layer of the $i$-th neighborhood aggregation layer.*

2. *GCN maps nodes to representations such that:*

$$\left\| \mathbf{h}_v^{(k)} - \mathbf{h}_u^{(k)} \right\|_2 \leq \prod_{i=1}^{k} L_f^{(i)} WD(\tilde{\mathcal{W}}_v^{(k)}, \tilde{\mathcal{W}}_u^{(k)})$$

   *where $L_f^{(i)}$ denotes the Lipschitz constant of the fully-connected layer of the $i$-th neighborhood aggregation layer.*

3. *Under mild assumptions (biases of MLPs are ignored), GIN-0 maps nodes to representations such that:*

$$\left\| \mathbf{h}_v^{(k)} - \mathbf{h}_u^{(k)} \right\|_2 \leq \prod_{i=1}^{k} L_f^{(i)} WD(\mathcal{W}_v^{(k)}, \mathcal{W}_u^{(k)})$$

   *where $L_f^{(i)}$ denotes the Lipschitz constant of the fully-connected layer of the $i$-th neighborhood aggregation layer.*

Once again, the above result highlights the limitations of the considered models. Specifically, our results imply that the representations of all models learned at the $k$-th neighborhood aggregation layer depend only on the nodes that can be reached in exactly $k$ steps and not on the intermediate nodes along each walk. Furthermore, we have bounded the Lipschitz constant of all the models with respect to the different types of walks starting from the different nodes.

We also empirically validate the theoretical analysis with experiments conducted on a synthetic dataset. As discussed above, the computational cost of solving the problem of equation 1 can become prohibitive. Therefore, we constructed a synthetic dataset where graphs are generated by applying perturbations to a perfectly balanced tree of height 2 with branching factor 10. More specifically, each node of the tree is initially annotated with a unique feature (a 7-dimensional binary vector). Then, nodes are randomly removed from the tree (and then any remaining disconnected nodes are also removed). Gaussian noise is then added to the feature vector of each node (zero mean and variance equal to $0.1$). When no normalization of the features takes place (as in the case of GIN-0), we can compute the WD of the roots of two trees since we know from construction which nodes of each tree match with the nodes of some other tree. We train the GIN-0 model on those trees in the task of predicting the number of nodes of each tree based on the root node's representation. We visualize the relation between the WD of two roots and their respective distance in the embedding

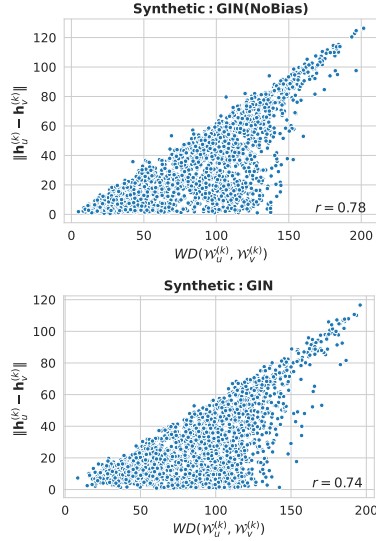

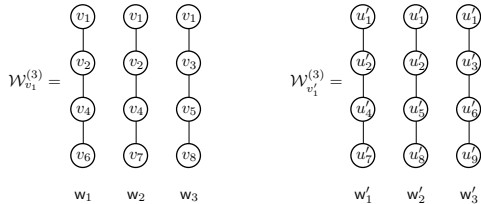

Figure 4: Example of two sets of walks $\mathcal{W}_{v_1}^{(3)} = \{(\mathsf{w}_1, \mathbf{h}_{\mathsf{w}_1}), (\mathsf{w}_2, \mathbf{h}_{\mathsf{w}_2}), (\mathsf{w}_3, \mathbf{h}_{\mathsf{w}_3})\}$ and $\mathcal{W}_{u_1'}^{(3)} = \{(\mathsf{w}_1', \mathbf{h}_{\mathsf{w}_1'}), (\mathsf{w}_2', \mathbf{h}_{\mathsf{w}_2'}), (\mathsf{w}_3', \mathbf{h}_{\mathsf{w}_3'})\}$ starting from nodes $v_1$ and $v_1'$, respectively. Suppose that $\mathbf{h}_{\mathsf{w}_1} = \mathbf{h}_{\mathsf{w}_1'}$, $\mathbf{h}_{\mathsf{w}_2} = \mathbf{h}_{\mathsf{w}_2'}$ and $\mathbf{h}_{\mathsf{w}_3} = \mathbf{h}_{\mathsf{w}_3'}$. Then, we can match walk $\mathsf{w}_3$ with walk $\mathsf{w}_3'$. If we also match walk $\mathsf{w}_1$ with walk $\mathsf{w}_1'$, then, we cannot match walk $\mathsf{w}_2$ with walk $\mathsf{w}_2'$ since $\mathrm{CSL}(\mathsf{w}_1, \mathsf{w}_2) = 3$ while $\mathrm{CSL}(\mathsf{w}_1', \mathsf{w}_2') = 2$. Thus, $\mathrm{WD}\big(\mathcal{W}_{v_1}^{(3)}, \mathcal{W}_{v_1'}^{(3)}\big) = ||\mathbf{h}_{\mathsf{w}_2}||_2 + ||\mathbf{h}_{\mathsf{w}_2'}||_2$.

Figure 3: Euclidean distances of the representations generated at the second layer of the GIN-0 model vs. WD of the nodes.

space (at the second message passing layer) in Figure 3. We observe that WD strongly correlates with the distance of the embeddings of the root nodes, supporting our theoretical results of defining the Lipschitz constant with respect to WD.

## 4.3 LINK TO OVERSQUASHING

Our theoretical results are also related to the phenomenon of oversquashing (Alon & Yahav, 2021; Topping et al., 2021) which occurs in MPNNs due to large information compression through bottlenecks. Specifically, messages that are propagated from distant nodes through certain bottlenecks of the graph, turn out to have negligible impact on the root node's representation. Our theoretical results suggest that given two nodes $v$ and $u$, the smaller the value of the fraction $w_{vu}^{(k)}/w_v^{(k)}$ (where $w_{vu}^{(k)}$ denotes the number of walks of length $k$ from node $v$ to node $u$), the less the impact of the message(s) from node $u$ to node $v$. To verify our claim, we constructed a graph classification task to investigate whether an MPNN model can capture the interaction between two nodes. All the generated graphs are instances of a single family of graphs. Specifically, each graph consists of two components: (1) a complete graph with $n$ nodes; and (2) a perfectly balanced $r$-ary tree of height 2. The two components are connected by an edge, between one of the nodes of the complete graph and the root of the tree. We use $\mathrm{CBT}(n, r)$ to denote such a graph with parameters $n$ and $r$. Figure 5 illustrates the $\mathrm{CBT}(4, 2)$ graph. Each graph belongs to one out of two classes. The class label depends on the features of the nodes. We annotate all nodes with an 8-dimensional feature vector. One of the nodes of the complete graph (not the one connected with the root of the tree) is annotated with a vector full of 5s and Gaussian noise is further added pointwise (zero mean and variance equal to 0.1). For graphs that belong to class 0, the rest of the nodes are annotated with vectors randomly sampled from the standard normal distribution. For graphs that belong to class 1, a single leaf of the tree is annotated with a vector full of $-5$s and Gaussian noise is further added pointwise (zero mean and variance equal to 0.1). We create three datasets in total where

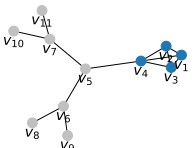

Figure 5: An example of the $\mathrm{CBT}(4, 2)$ graph.

in each case $n$ and $r$ take the following values: (1) $\mathrm{CBT}(5 - 10, 5 - 8)$ where $n \in \{5, 6, \ldots, 10\}$ and $r \in \{5, 6, 7, 8\}$; (2) $\mathrm{CBT}(2 - 3, 1 - 5)$ where $n \in \{2, 3\}$ and $r \in \{1, 2, \ldots, 5\}$; and (3) $\mathrm{CBT}(5 - 20, 1)$ where $n \in \{5, 6, \ldots, 20\}$ and $r \in \{1\}$. For each dataset, we construct 200 graphs. Graphs of each class are generated with equal probability. We split each dataset into a training set

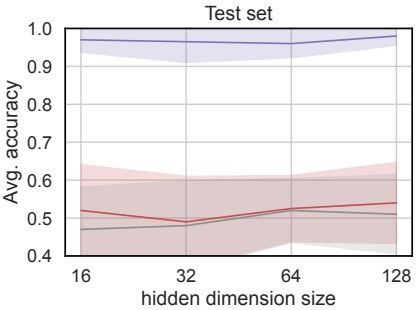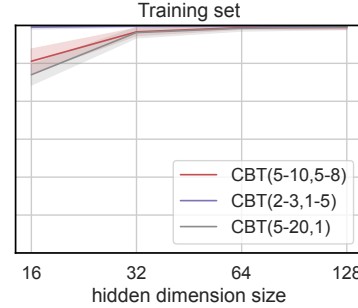

Figure 6: Average training and test accuracies achieved by the GIN-0 model on the three generated datasets.

(80%), a validation set (10%), and a test set (10%), and train the GIN-0 model on each of these datasets. We set the number of neighborhood aggregation layers to 4 (i.e., shortest path distance between nodes that interact with each other). To update the node features, we use in each neighborhood aggregation layer a multi-layer perceptron that consists of 2 fully connected layers. Each fully connected layer is followed by the ReLU activation function, while batch normalization is applied to the node representations that are produced by the first fully connected layer. To make a prediction, we feed the representation of the node of the complete graph that is annotated with a vector full of 5s (along with noise) to a fully connected layer. We set the hidden dimension size to 16, 32, 64, and 128 and provide different results for each one of these values. We train the model for 300 epochs and use the model that achieved the lowest loss on the validation set to make predictions for the test samples. We repeat each experiment 10 times and we report the average accuracies on the training and test sets. The results are provided in Figure 6. Let $v$ and $u$ be the node of the complete graph and the node of the tree that interact with each other in the graphs that belong to class 1. We observe that the model achieves very high levels of performance on both the training and test sets of the CBT$(2 - 3, 1 - 5)$ dataset. Note that in these graphs, the number of walks $w_v^{(4)}$ of length 4 that start from node $v$ is not very large since the complete graph consists only of 2 or 3 nodes. The model also achieves average accuracies close to 1 on the training sets of the other two datasets (i.e., CBT$(5 - 10, 5 - 8)$ and CBT$(5 - 20, 1)$), however, on the respective test sets, the accuracies are very close to 0.5 which demonstrates that the model cannot capture the interaction between nodes $v$ and $u$. The increase in the hidden dimension size does not seem to improve the model's performance. In the graphs that are contained in both these datasets, the number of walks $w_v^{(4)}$ of length 4 that start from node $v$ is much larger than the respective number in the graphs contained in the CBT$(2 - 3, 1 - 5)$ dataset. We need to mention that, in contrast to previous studies which claim that oversquashing is a result of bottlenecks that exist in the graph, our results suggest that it is mainly because of the large number of walks that originate at the different nodes. This is clearly evidenced in our empirical findings since no bottlenecks exist between nodes $v$ and $u$ in the graphs of the CBT$(5 - 20, 1)$ dataset, but still, GIN-0 was unable to encode their interaction into $v$'s learned representation. This result redefines our understanding of oversquashing and its implications in the context of graph-based learning.

## 5 CONCLUSION

In this paper, we focused on four well-established GNN models and we investigated what properties of graphs these models can capture. First, we considered the case where no node features are available and nodes are annotated with the same features. We found that two of the models capture no structural properties of graphs since they embed all nodes into the same feature vector. We further show that nodes with divergent structural characteristics can have similar representations when they share similar $k$-length walk patterns. Finally, we showed that oversquashing is not exclusively a product of bottlenecks within the graph structure. Instead, it predominantly arises when the number of walks from one node to another is disproportionally small compared to the total walks originating from the latter, regardless of bottleneck presence.

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

## A   PROOF OF THEOREM 4.1

We assume that all the nodes of the graph are initially annotated with a single feature equal to $1$.

### A.1   DGCNN

The DGCNN model that updates node representations as follows (Zhang et al., 2018):

$$\mathbf{h}_v^{(k)} = f\left( \sum_{u \in \mathcal{N}(v) \cup \{v\}} \frac{1}{\deg(v) + 1} \mathbf{W}^{(k)} \mathbf{h}_u^{(k-1)} \right)$$

Once again, we assume that all the nodes of the graph are initially annotated with a single feature equal to $1$. We will show by induction that the model maps all nodes to the same vector representation. We first assume that $\mathbf{h}_v^{k-1} = \mathbf{h}_u^{k-1} = \mathbf{h}^{k-1}$ for all $v, u \in V$. This is actually true for $k = 1$ since $\mathbf{h}_v^0 = 1$ for all $v \in V$. Then, we have that $\mathbf{W}^{(k)}\mathbf{h}_v^{(k-1)} = \mathbf{W}^{(k)}\mathbf{h}_u^{(k-1)} = \mathbf{W}^{(k)}\mathbf{h}^{(k-1)}$ for all $v, u \in V$. We also have that:

$$\sum_{u \in \mathcal{N}(v) \cup \{v\}} \frac{1}{\deg(v) + 1} = 1$$

Thus, we finally have that:

$$\mathbf{h}_v^{(k)} = f\left( \sum_{u \in \mathcal{N}(v) \cup \{v\}} \frac{1}{\deg(v) + 1} \mathbf{W}^{(k)} \mathbf{h}_u^{(k-1)} \right) = f\left( \mathbf{W}^{(k)} \mathbf{h}^{(k-1)} \right)$$

for all $v \in V$. We have shown that this variant of the GCN model produces the same representation for all nodes of all graphs and thus, it cannot capture any structural information about the neighborhood of each node.

## A.2 GAT

The GAT model updates node representations as follows:

$$\mathbf{h}_v^{(k)} = \sigma \left( \sum_{u \in \mathcal{N}(v)} \alpha_{vu} \mathbf{W}^{(k)} \mathbf{h}_u^{(k-1)} \right)$$

where $\alpha_{vu}$ is an attention coefficient that indicates the importance of node $u$'s features to node $v$. Once again, we assume that all the nodes of the graph are initially annotated with a single feature equal to $1$. We will show by induction that the model maps all nodes to the same vector representation. We first assume that $\mathbf{h}_v^{k-1} = \mathbf{h}_u^{k-1} = \mathbf{h}^{k-1}$ for all $v, u \in V$. This is actually true for $k = 1$ since $\mathbf{h}_v^0 = 1$ for all $v \in V$. Then, we have that $\mathbf{W}^{(k)} \mathbf{h}_v^{(k-1)} = \mathbf{W}^{(k)} \mathbf{h}_u^{(k-1)} = \mathbf{W}^{(k)} \mathbf{h}^{(k-1)}$ for all $v, u \in V$. Since the attention coefficients are normalized, we have:

$$\sum_{u \in \mathcal{N}(v)} \alpha_{vu} = 1$$

Thus, we finally have that:

$$\mathbf{h}_v^{(k)} = \sigma \left( \sum_{u \in \mathcal{N}(v)} \alpha_{vu} \mathbf{W}^{(k)} \mathbf{h}_u^{(k-1)} \right) = \sigma \left( \mathbf{W}^{(k)} \mathbf{h}^{(k-1)} \right)$$

for all $v \in V$. We have shown that the GAT model produces the same representation for all nodes of all graphs and thus, it cannot capture any structural information about the neighborhood of each node.

## A.3 GCN

The GCN model updates node representations as follows:

$$\mathbf{h}_v^{(k)} = \mathrm{ReLU} \left( \sum_{u \in \mathcal{N}(v) \cup \{v\}} \frac{\mathbf{W}^{(k)} \mathbf{h}_u^{(k-1)}}{\sqrt{(1 + \deg(v))(1 + \deg(u))}} \right)$$

Note that the GCN model (as decribed in the original paper (Kipf & Welling, 2017)) does not perform an affine transformation of the node features, but instead a linear transformation. In other words, no biases are present. Thus, the following holds:

$$\mathbf{h}_v^{(k)} = \mathrm{ReLU} \left( \sum_{u \in \mathcal{N}(v) \cup \{v\}} \frac{\mathbf{W}^{(k)} \mathbf{h}_u^{(k-1)}}{\sqrt{(1 + \deg(v))(1 + \deg(u))}} \right) = \mathrm{ReLU} \left( \mathbf{W}^{(k)} \sum_{u \in \mathcal{N}(v) \cup \{v\}} \frac{\mathbf{h}_u^{(k-1)}}{\sqrt{(1 + \deg(v))(1 + \deg(u))}} \right)$$

$$(2)$$

Let $L_f^{(k)}$ denote the Lipschitz constant associated with the fully connected layer of the $k$-th neighborhood aggregation layer of GCN. In what follows, we set $d(v) = 1 + \deg(v)$ and also denote the set $\mathcal{N}(v) \cup \{v\}$ by $\tilde{\mathcal{N}}(v)$. Then, we have:

$$||\mathbf{h}_v^{(1)} - \mathbf{h}_{v'}^{(1)}||_2 = \left\| \mathrm{ReLU} \left( \sum_{u \in \tilde{\mathcal{N}}(v)} \frac{\mathbf{W}^{(1)} \mathbf{h}_u^{(0)}}{\sqrt{d(v)d(u)}} \right) - \mathrm{ReLU} \left( \sum_{u' \in \tilde{\mathcal{N}}(v')} \frac{\mathbf{W}^{(1)} \mathbf{h}_{u'}^{(0)}}{\sqrt{d(v')d(u')}} \right) \right\|_2$$

$$= \left\| \mathrm{ReLU} \left( \mathbf{W}^{(1)} \sum_{u \in \tilde{\mathcal{N}}(v)} \frac{\mathbf{h}_u^{(0)}}{\sqrt{d(v)d(u)}} \right) - \mathrm{ReLU} \left( \mathbf{W}^{(1)} \sum_{u' \in \tilde{\mathcal{N}}(v')} \frac{\mathbf{h}_{u'}^{(0)}}{\sqrt{d(v')d(u')}} \right) \right\|_2$$

$$\leq L_f^{(1)} \left\| \sum_{u \in \tilde{\mathcal{N}}(v)} \frac{1}{\sqrt{d(v)d(u)}} - \sum_{u' \in \tilde{\mathcal{N}}(v')} \frac{1}{\sqrt{d(v')d(u')}} \right\|_2$$

$$= L_f^{(1)}\left\|\tilde{w}_v^{(1)} - \tilde{w}_{v'}^{(1)}\right\|_2$$

$$\|\mathbf{h}_v^{(2)} - \mathbf{h}_{v'}^{(2)}\|_2 = \left\|\text{ReLU}\left(\sum_{u\in\tilde{\mathcal{N}}(v)} \frac{\mathbf{W}^{(2)}\mathbf{h}_u^{(1)}}{\sqrt{d(v)d(u)}}\right) - \text{ReLU}\left(\sum_{u'\in\tilde{\mathcal{N}}(v')} \frac{\mathbf{W}^{(2)}\mathbf{h}_{u'}^{(1)}}{\sqrt{d(v')d(u')}}\right)\right\|_2$$

$$= \left\|\text{ReLU}\left(\mathbf{W}^{(2)}\sum_{u\in\tilde{\mathcal{N}}(v)} \frac{\mathbf{h}_u^{(1)}}{\sqrt{d(v)d(u)}}\right) - \text{ReLU}\left(\mathbf{W}^{(2)}\sum_{u'\in\tilde{\mathcal{N}}(v')} \frac{\mathbf{h}_{u'}^{(1)}}{\sqrt{d(v')d(u')}}\right)\right\|_2$$

$$\leq L_f^{(2)}\left\|\sum_{u\in\tilde{\mathcal{N}}(v)} \frac{1}{\sqrt{d(v)d(u)}}\text{ReLU}\left(\sum_{w\in\tilde{\mathcal{N}}(u)} \frac{\mathbf{W}^{(1)}\mathbf{h}_w^{(0)}}{\sqrt{d(u)d(w)}}\right)\right.$$
$$\left. - \sum_{u'\in\tilde{\mathcal{N}}(v')} \frac{1}{\sqrt{d(v')d(u')}}\text{ReLU}\left(\sum_{w'\in\tilde{\mathcal{N}}(u')} \frac{\mathbf{W}^{(1)}\mathbf{h}_{w'}^{(0)}}{\sqrt{d(u')d(w')}}\right)\right\|_2$$

$$= L_f^{(2)}\left\|\sum_{u\in\tilde{\mathcal{N}}(v)} \frac{1}{\sqrt{d(v)d(u)}}\text{ReLU}\left(\mathbf{W}^{(1)}\sum_{w\in\tilde{\mathcal{N}}(u)} \frac{\mathbf{h}_w^{(0)}}{\sqrt{d(u)d(w)}}\right)\right.$$
$$\left. - \sum_{u'\in\tilde{\mathcal{N}}(v')} \frac{1}{\sqrt{d(v')d(u')}}\text{ReLU}\left(\mathbf{W}^{(1)}\sum_{w'\in\tilde{\mathcal{N}}(u')} \frac{\mathbf{h}_{w'}^{(0)}}{\sqrt{d(u')d(w')}}\right)\right\|_2$$

$$\leq L_f^{(2)}L_f^{(1)}\left\|\sum_{u\in\tilde{\mathcal{N}}(v)} \frac{1}{\sqrt{d(v)d(u)}}\sum_{w\in\tilde{\mathcal{N}}(u)} \frac{1}{\sqrt{d(u)d(w)}} - \sum_{u'\in\tilde{\mathcal{N}}(v')} \frac{1}{\sqrt{d(v')d(u')}}\sum_{w'\in\tilde{\mathcal{N}}(u')} \frac{1}{\sqrt{d(u')d(w')}}\right\|_2$$

$$= L_f^{(2)}L_f^{(1)}\left\|\sum_{u\in\tilde{\mathcal{N}}(v)}\sum_{w\in\tilde{\mathcal{N}}(u)} \frac{1}{d(u)\sqrt{d(v)d(w)}} - \sum_{u'\in\tilde{\mathcal{N}}(v')}\sum_{w'\in\tilde{\mathcal{N}}(u')} \frac{1}{d(u')\sqrt{d(v')d(w')}}\right\|_2$$

$$= L_f^{(2)}L_f^{(1)}\left\|\tilde{w}_v^{(2)} - \tilde{w}_{v'}^{(2)}\right\|_2$$

$$\vdots$$

$$\|\mathbf{h}_v^{(K)} - \mathbf{h}_{v'}^{(K)}\|_2 = \left\|\text{ReLU}\left(\sum_{u\in\tilde{\mathcal{N}}(v)} \frac{\mathbf{W}^{(K)}\mathbf{h}_u^{(K-1)}}{\sqrt{d(v)d(u)}}\right) - \text{ReLU}\left(\sum_{u'\in\tilde{\mathcal{N}}(v')} \frac{\mathbf{W}^{(K)}\mathbf{h}_{u'}^{(K-1)}}{\sqrt{d(v')d(u')}}\right)\right\|_2$$

$$= \left\|\text{ReLU}\left(\mathbf{W}^{(K)}\sum_{u\in\tilde{\mathcal{N}}(v)} \frac{\mathbf{h}_u^{(K-1)}}{\sqrt{d(v)d(u)}}\right) - \text{ReLU}\left(\mathbf{W}^{(K)}\sum_{u'\in\tilde{\mathcal{N}}(v')} \frac{\mathbf{h}_{u'}^{(K-1)}}{\sqrt{d(v')d(u')}}\right)\right\|_2$$

$$\leq L_f^{(K)}\left\|\sum_{u\in\tilde{\mathcal{N}}(v)} \frac{1}{\sqrt{d(v)d(u)}}\text{ReLU}\left(\sum_{w\in\tilde{\mathcal{N}}(u)} \frac{\mathbf{W}^{(K-1)}\mathbf{h}_w^{(K-2)}}{\sqrt{d(u)d(w)}}\right)\right.$$
$$\left. - \sum_{u'\in\tilde{\mathcal{N}}(v')} \frac{1}{\sqrt{d(v')d(u')}}\text{ReLU}\left(\sum_{w'\in\tilde{\mathcal{N}}(u')} \frac{\mathbf{W}^{(K-1)}\mathbf{h}_{w'}^{(K-2)}}{\sqrt{d(u')d(w')}}\right)\right\|_2$$

$$= L_f^{(K)}\left\|\sum_{u\in\tilde{\mathcal{N}}(v)} \frac{1}{\sqrt{d(v)d(u)}}\text{ReLU}\left(\mathbf{W}^{(K-1)}\sum_{w\in\tilde{\mathcal{N}}(u)} \frac{\mathbf{h}_w^{(K-2)}}{\sqrt{d(u)d(w)}}\right)\right.$$
$$\left. - \sum_{u'\in\tilde{\mathcal{N}}(v')} \frac{1}{\sqrt{d(v')d(u')}}\text{ReLU}\left(\mathbf{W}^{(K-1)}\sum_{w'\in\tilde{\mathcal{N}}(u')} \frac{\mathbf{h}_{w'}^{(K-2)}}{\sqrt{d(u')d(w')}}\right)\right\|_2$$

$$\leq L_f^{(K)}L_f^{(K-1)}\cdots L_f^{(2)}\left\|\sum_{u\in\tilde{\mathcal{N}}(v)} \frac{1}{\sqrt{d(v)d(u)}}\sum_{w\in\tilde{\mathcal{N}}(u)} \frac{1}{\sqrt{d(u)d(w)}}\cdots\text{ReLU}\left(\sum_{a\in\tilde{\mathcal{N}}(b)} \frac{\mathbf{W}^{(1)}\mathbf{h}_a^{(0)}}{\sqrt{d(b)d(a)}}\right)\right.$$

$$- \sum_{u' \in \tilde{\mathcal{N}}(v')} \frac{1}{\sqrt{d(v')d(u')}} \sum_{w' \in \tilde{\mathcal{N}}(u')} \frac{1}{\sqrt{d(u')d(w')}} \dots \text{ReLU}\left( \sum_{a' \in \tilde{\mathcal{N}}(b')} \frac{\mathbf{W}^{(1)}\mathbf{h}_{a'}^{(0)}}{\sqrt{d(b')d(a')}} \right) \Bigg\|_2$$

$$= L_f^{(K)} L_f^{(K-1)} \dots L_f^{(2)} \Bigg\| \sum_{u \in \tilde{\mathcal{N}}(v)} \frac{1}{\sqrt{d(v)d(u)}} \sum_{w \in \tilde{\mathcal{N}}(u)} \frac{1}{\sqrt{d(u)d(w)}} \dots \text{ReLU}\left( \mathbf{W}^{(1)} \sum_{a \in \tilde{\mathcal{N}}(b)} \frac{\mathbf{h}_a^{(0)}}{\sqrt{d(b)d(a)}} \right)$$

$$- \sum_{u' \in \tilde{\mathcal{N}}(v')} \frac{1}{\sqrt{d(v')d(u')}} \sum_{w' \in \tilde{\mathcal{N}}(u')} \frac{1}{\sqrt{d(u')d(w')}} \dots \text{ReLU}\left( \mathbf{W}^{(1)} \sum_{a' \in \tilde{\mathcal{N}}(b')} \frac{\mathbf{h}_{a'}^{(0)}}{\sqrt{d(b')d(a')}} \right) \Bigg\|_2$$

$$\leq L_f^{(K)} L_f^{(K-1)} \dots L_f^{(1)} \Bigg\| \sum_{u \in \tilde{\mathcal{N}}(v)} \frac{1}{\sqrt{d(v)d(u)}} \sum_{w \in \tilde{\mathcal{N}}(u)} \frac{1}{\sqrt{d(u)d(w)}} \dots \sum_{a \in \tilde{\mathcal{N}}(b)} \frac{1}{\sqrt{d(b)d(a)}}$$

$$- \sum_{u' \in \tilde{\mathcal{N}}(v')} \frac{1}{\sqrt{d(v')d(u')}} \sum_{w' \in \tilde{\mathcal{N}}(u')} \frac{1}{\sqrt{d(u')d(w')}} \dots \sum_{a' \in \tilde{\mathcal{N}}(b')} \frac{1}{\sqrt{d(b')d(a')}} \Bigg\|_2$$

$$= L_f^{(K)} L_f^{(K-1)} \dots L_f^{(1)} \Bigg\| \sum_{u \in \tilde{\mathcal{N}}(v)} \sum_{w \in \tilde{\mathcal{N}}(u)} \dots \sum_{a \in \tilde{\mathcal{N}}(b)} \frac{1}{d(u)d(w)\dots d(b)} \frac{1}{\sqrt{d(v)d(a)}}$$

$$- \sum_{u' \in \tilde{\mathcal{N}}(v')} \sum_{w' \in \tilde{\mathcal{N}}(u')} \dots \sum_{a' \in \tilde{\mathcal{N}}(b')} \frac{1}{d(u')d(w')\dots d(b')} \frac{1}{\sqrt{d(v')d(a')}} \Bigg\|_2$$

$$= L_f^{(K)} L_f^{(K-1)} \dots L_f^{(1)} \left\| \tilde{w}_v^{(K)} - \tilde{w}_{v'}^{(K)} \right\|_2$$

It turns out that the node representations learned at the $k$-th layer of a GCN model are related to the normalized number of walks of length $k$ starting from each node. Given a walk of length $k$ consisting of the following nodes $(v_1, v_2, \dots, v_k)$, the walk is normalized by the product of the degrees of nodes $v_2, \dots, v_{k-1}$ each increased by 1 and of the square root of the degrees of nodes $v_1$ and $v_k$ also increased by 1. Thus, the contribution of each walk is inversely proportional to the degrees of the nodes of which the walk is composed.

## A.4  GIN-0

The GIN-0 model updates node representations as follows:

$$\mathbf{h}_v^{(k)} = \text{MLP}^{(k)}\left( \left( 1 + \epsilon^{(k)} \right) \mathbf{h}_v^{(k-1)} + \sum_{u \in \mathcal{N}(v)} \mathbf{h}_u^{(k-1)} \right) = \text{MLP}^{(k)}\left( \sum_{u \in \mathcal{N}(v) \cup \{v\}} \mathbf{h}_u^{(k-1)} \right) \quad (3)$$

We make the following assumption.

**Assumption A.1.** *We assume that the biases of the fully-connected layers of all MLPs are equal to zero vectors.*

Our results are also valid when given fully-connected layers of the form $\text{fc}(\mathbf{x}) = \mathbf{W}\mathbf{x} + \mathbf{b}$, the following holds $||\mathbf{b}|| \ll ||\mathbf{W}\mathbf{x}||$. This is a reasonable assumption which is common in real-world machine learning settings. Note that the activation function of the MLPs of the GIN-0 model is the ReLU function (Xu et al., 2019). Without loss of generality, we assume that the MLPs consist of two fully-connected layers. Given the above, the update function of the GIN-0 model takes the following form:

$$\text{MLP}^{(k)}(\mathbf{x}) = \text{ReLU}\left( \mathbf{W}_2^{(k)} \text{ReLU}\left( \mathbf{W}_1^{(k)} \mathbf{x} \right) \right) \quad (4)$$

We next prove the following Lemma which will be useful in our analysis.

**Lemma A.2.** *Let MLP denote the model defined in Equation equation 4 above. Let also $\mathcal{X} = \{\mathbf{x}_1, \dots, \mathbf{x}_M\}$ denote a set of vectors such that given any two vectors from the set, one is a scalar*

*multiple of the other, i. e., if $\mathbf{x}_i, \mathbf{x}_j \in \mathcal{X}$, then $\mathbf{x}_i = a\mathbf{x}_j$ where $a > 0$. Then, the following holds:*

$$\sum_{i=1}^{M} MLP(\mathbf{x}_i) = MLP\left(\sum_{i=1}^{M} \mathbf{x}_i\right)$$

*Proof.* For $a > 0$, we have that $\mathbf{W}(a\mathbf{x}) = a\mathbf{W}\mathbf{x}$. Furthermore, we also have that $\mathrm{ReLU}(a\mathbf{x}) = a\mathrm{ReLU}(\mathbf{x})$. Then, we have:

$$\mathrm{MLP}(a\mathbf{x}) = \mathrm{ReLU}\left(\mathbf{W}_2 \mathrm{ReLU}\left(\mathbf{W}_1(a\mathbf{x})\right)\right) = a\mathrm{ReLU}\left(\mathbf{W}_2 \mathrm{ReLU}\left(\mathbf{W}_1(\mathbf{x})\right)\right) = a\mathrm{MLP}(\mathbf{x})$$

We have assumed that $\mathbf{x}_2 = a_2\mathbf{x}_1, \mathbf{x}_3 = a_3\mathbf{x}_1, \ldots, \mathbf{x}_M = a_M\mathbf{x}_1$. Then, we have that:

$$\begin{aligned}
\sum_{i=1}^{M} \mathrm{MLP}(\mathbf{x}_i) &= \mathrm{MLP}(\mathbf{x}_1) + \sum_{i=2}^{M} \mathrm{MLP}(a_i\mathbf{x}_1) \\
&= \mathrm{MLP}(\mathbf{x}_1) + \sum_{i=2}^{M} a_i\mathrm{MLP}(\mathbf{x}_1) \\
&= (1 + a_2 + \ldots + a_M)\mathrm{MLP}(\mathbf{x}_1) \\
&= \mathrm{MLP}\left((1 + a_2 + \ldots + a_M)\mathbf{x}_1\right) \\
&= \mathrm{MLP}\left(\sum_{i=1}^{M} \mathbf{x}_i\right)
\end{aligned}$$

$\square$

It is trivial to generalize the above Lemma to MLPs that contain more than two layers. We also prove the following Lemma.

**Lemma A.3.** *Let the MLPs of the GIN-$0$ model be instances of the MLP of Equation equation 4 above. Let $\mathcal{V}$ denote the set of nodes of all graphs and let $\mathcal{X}^{(k-1)} = \{\!\{\mathbf{h}_1^{(k-1)}, \ldots, \mathbf{h}_{|\mathcal{V}|}^{(k-1)}\}\!\}$ be the multiset of node representations that emerged at the $(k-1)$-th layer of the model. Suppose that given any two vectors from this multiset, one is a scalar multiple of the other, i. e., if $\mathbf{h}_i^{(k-1)}, \mathbf{h}_j^{(k-1)} \in \mathcal{X}^{(k-1)}$, then $\mathbf{h}_i^{(k-1)} = a\mathbf{h}_j^{(k-1)}$ where $a > 0$. Then, the same holds for the node representations that emerge at the $k$-th layer of the model, i. e., for any two vectors $\mathbf{h}_i^{(k)}, \mathbf{h}_j^{(k)} \in \mathcal{X}^{(k)} = \{\!\{\mathbf{h}_1^{(k)}, \ldots, \mathbf{h}_{|\mathcal{V}|}^{(k)}\}\!\}$, we have that $\mathbf{h}_i^{(k)} = a\mathbf{h}_j^{(k)}$ where $a > 0$.*

*Proof.* For $a > 0$, we have that $\mathbf{W}(a\mathbf{x}) = a\mathbf{W}\mathbf{x}$. Furthermore, we also have that $\mathrm{ReLU}(a\mathbf{x}) = a\mathrm{ReLU}(\mathbf{x})$. Then, we have:

$$\mathrm{MLP}(a\mathbf{x}) = \mathrm{ReLU}\left(\mathbf{W}_2 \mathrm{ReLU}\left(\mathbf{W}_1(a\mathbf{x})\right)\right) = a\mathrm{ReLU}\left(\mathbf{W}_2 \mathrm{ReLU}\left(\mathbf{W}_1(\mathbf{x})\right)\right) = a\mathrm{MLP}(\mathbf{x})$$

We have assumed that given some node $w \in \mathcal{V}$, then for any node $u \in \mathcal{V}$, we have that $\mathbf{h}_u^{(k-1)} = a_u\mathbf{h}_w^{(k-1)}$. Then, given any node $v \in \mathcal{V}$, its representation is updated as follows:

$$\begin{aligned}
\mathbf{h}_v^{(k)} = \mathrm{MLP}^{(k)}\left(\sum_{u \in \mathcal{N}(v) \cup \{v\}} \mathbf{h}_u^{(k-1)}\right) &= \mathrm{MLP}^{(k)}\left(\sum_{u \in \mathcal{N}(v) \cup \{v\}} a_u\mathbf{h}_w^{(k-1)}\right) \\
&= \sum_{u \in \mathcal{N}(v) \cup \{v\}} a_u\mathrm{MLP}^{(k)}\left(\mathbf{h}_w^{(k-1)}\right) \\
&= c\mathrm{MLP}^{(k)}\left(\mathbf{h}_w^{(k-1)}\right)
\end{aligned}$$

which concludes the proof. $\square$

Thus, the above Lemma suggests that for MLPs of the form of Equation equation 4, the node representations produced by GIN-0 are either scalar multiples of each other and they point in the same direction or are all equal to the all-zeros vector.

Let $L_f^{(k)}$ denote the Lipschitz constant associated with the MLP of the $k$-th neighborhood aggregation layer of GIN-0. We assume that all the nodes of the graph are initially annotated with a single feature equal to 1. We also denote the set $\mathcal{N}(v) \cup \{v\}$ by $\tilde{\mathcal{N}}(v)$. Then, we have:

$$
\begin{aligned}
||\mathbf{h}_v^{(1)} - \mathbf{h}_{v'}^{(1)}||_2 &= \left\| \mathrm{MLP}^{(1)}\left( \sum_{u\in\tilde{\mathcal{N}}(v)} \mathbf{h}_u^{(0)} \right) - \mathrm{MLP}^{(1)}\left( \sum_{u'\in\tilde{\mathcal{N}}(v')} \mathbf{h}_{u'}^{(0)} \right) \right\|_2 \\
&\leq L_f^{(1)} \left\| \sum_{u\in\tilde{\mathcal{N}}(v)} 1 - \sum_{u'\in\tilde{\mathcal{N}}(v')} 1 \right\|_2 \\
&= L_f^{(1)} \left\| (\deg(v)+1) - (\deg(v')+1) \right\|_2 \\
&= L_f^{(1)} \left\| w_v^{(1)} - w_{v'}^{(1)} \right\|_2
\end{aligned}
$$

$$
\begin{aligned}
||\mathbf{h}_v^{(2)} - \mathbf{h}_{v'}^{(2)}||_2 &= \left\| \mathrm{MLP}^{(2)}\left( \sum_{u\in\tilde{\mathcal{N}}(v)} \mathbf{h}_u^{(1)} \right) - \mathrm{MLP}^{(2)}\left( \sum_{u'\in\tilde{\mathcal{N}}(v')} \mathbf{h}_{u'}^{(1)} \right) \right\|_2 \\
&\leq L_f^{(2)} \left\| \sum_{u\in\tilde{\mathcal{N}}(v)} \mathrm{MLP}^{(1)}\left( \sum_{w\in\tilde{\mathcal{N}}(u)} \mathbf{h}_w^{(0)} \right) - \sum_{u'\in\tilde{\mathcal{N}}(v')} \mathrm{MLP}^{(1)}\left( \sum_{w'\in\tilde{\mathcal{N}}(u')} \mathbf{h}_{w'}^{(0)} \right) \right\|_2 \\
&= L_f^{(2)} \left\| \mathrm{MLP}^{(1)}\left( \sum_{u\in\tilde{\mathcal{N}}(v)} \sum_{w\in\tilde{\mathcal{N}}(u)} \mathbf{h}_w^{(0)} \right) - \mathrm{MLP}^{(1)}\left( \sum_{u'\in\tilde{\mathcal{N}}(v')} \sum_{w'\in\tilde{\mathcal{N}}(u')} \mathbf{h}_{w'}^{(0)} \right) \right\|_2 \quad \text{(Lemma A.2)} \\
&\leq L_f^{(2)} L_f^{(1)} \left\| \sum_{u\in\tilde{\mathcal{N}}(v)} \sum_{w\in\tilde{\mathcal{N}}(u)} \mathbf{h}_w^{(0)} - \sum_{u'\in\tilde{\mathcal{N}}(v')} \sum_{w'\in\tilde{\mathcal{N}}(u')} \mathbf{h}_{w'}^{(0)} \right\|_2 \\
&= L_f^{(2)} L_f^{(1)} \left\| \sum_{u\in\tilde{\mathcal{N}}(v)} \sum_{w\in\tilde{\mathcal{N}}(u)} 1 - \sum_{u'\in\tilde{\mathcal{N}}(v')} \sum_{w'\in\tilde{\mathcal{N}}(u')} 1 \right\|_2 \\
&= L_f^{(2)} L_f^{(1)} \left\| w_v^{(2)} - w_{v'}^{(2)} \right\|_2
\end{aligned}
$$

$$\vdots$$

$$
\begin{aligned}
||\mathbf{h}_v^{(K)} - \mathbf{h}_{v'}^{(K)}||_2 &= \left\| \mathrm{MLP}^{(K)}\left( \sum_{u\in\tilde{\mathcal{N}}(v)} \mathbf{h}_u^{(K-1)} \right) - \mathrm{MLP}^{(K)}\left( \sum_{u'\in\tilde{\mathcal{N}}(v')} \mathbf{h}_{u'}^{(K-1)} \right) \right\|_2 \\
&\leq L_f^{(K)} \left\| \sum_{u\in\tilde{\mathcal{N}}(v)} \mathrm{MLP}^{(K-1)}\left( \sum_{w\in\tilde{\mathcal{N}}(u)} \mathbf{h}_w^{(K-2)} \right) - \sum_{u'\in\tilde{\mathcal{N}}(v')} \mathrm{MLP}^{(K-1)}\left( \sum_{w'\in\tilde{\mathcal{N}}(u')} \mathbf{h}_{w'}^{(K-2)} \right) \right\|_2 \\
&= L_f^{(K)} \left\| \mathrm{MLP}^{(K-1)}\left( \sum_{u\in\tilde{\mathcal{N}}(v)} \sum_{w\in\tilde{\mathcal{N}}(u)} \mathbf{h}_w^{(K-2)} \right) - \mathrm{MLP}^{(K-1)}\left( \sum_{u'\in\tilde{\mathcal{N}}(v')} \sum_{w'\in\tilde{\mathcal{N}}(u')} \mathbf{h}_{w'}^{(K-2)} \right) \right\|_2 \quad \text{(Lemma A.2)} \\
&\leq L_f^{(K)} L_f^{(K-1)} \cdots L_f^{(2)} \left\| \sum_{u\in\tilde{\mathcal{N}}(v)} \sum_{w\in\tilde{\mathcal{N}}(u)} \cdots \mathrm{MLP}^{(1)}\left( \sum_{a\in\tilde{\mathcal{N}}(b)} \mathbf{h}_a^{(0)} \right) \right. \\
&\qquad\qquad \left. - \sum_{u'\in\tilde{\mathcal{N}}(v')} \sum_{w'\in\tilde{\mathcal{N}}(u')} \cdots \mathrm{MLP}^{(1)}\left( \sum_{a'\in\tilde{\mathcal{N}}(b')} \mathbf{h}_{a'}^{(0)} \right) \right\|_2
\end{aligned}
$$

$$= L_f^{(K)} L_f^{(K-1)} \cdots L_f^{(2)} \left\| \mathrm{MLP}^{(1)} \left( \sum_{u \in \tilde{\mathcal{N}}(v)} \sum_{w \in \tilde{\mathcal{N}}(u)} \cdots \sum_{a \in \tilde{\mathcal{N}}(b)} \mathbf{h}_a^{(0)} \right) \right.$$

$$\left. - \mathrm{MLP}^{(1)} \left( \sum_{u' \in \tilde{\mathcal{N}}(v')} \sum_{w' \in \tilde{\mathcal{N}}(u')} \cdots \sum_{a' \in \tilde{\mathcal{N}}(b')} \mathbf{h}_{a'}^{(0)} \right) \right\|_2 \quad (\text{Lemma A.2})$$

$$\leq L_f^{(K)} L_f^{(K-1)} \cdots L_f^{(1)} \left\| \sum_{u \in \tilde{\mathcal{N}}(v)} \sum_{w \in \tilde{\mathcal{N}}(u)} \cdots \sum_{a \in \tilde{\mathcal{N}}(b)} \mathbf{h}_a^{(0)} - \sum_{u' \in \tilde{\mathcal{N}}(v')} \sum_{w' \in \tilde{\mathcal{N}}(u')} \cdots \sum_{a' \in \tilde{\mathcal{N}}(b')} \mathbf{h}_{a'}^{(0)} \right\|_2$$

$$= L_f^{(K)} L_f^{(K-1)} \cdots L_f^{(1)} \left\| \sum_{u \in \tilde{\mathcal{N}}(v)} \sum_{w \in \tilde{\mathcal{N}}(u)} \cdots \sum_{a \in \tilde{\mathcal{N}}(b)} 1 - \sum_{u' \in \tilde{\mathcal{N}}(v')} \sum_{w' \in \tilde{\mathcal{N}}(u')} \cdots \sum_{a' \in \tilde{\mathcal{N}}(b')} 1 \right\|_2$$

$$= L_f^{(K)} L_f^{(K-1)} \cdots L_f^{(2)} \left\| w_v^{(K)} - w_{v'}^{(K)} \right\|_2$$

It turns out that for $k = 1$, $\deg(v)$ and $\deg(u)$ are equal to the number of walks of length 1 starting from nodes $v$ and $u$, respectively. Likewise, for $k = 2$, $\sum_{w \in \mathcal{N}(v) \cup \{v\}} \sum_{x \in \mathcal{N}(w) \cup \{w\}} 1$ and $\sum_{w \in \mathcal{N}(u) \cup \{u\}} \sum_{x \in \mathcal{N}(w) \cup \{w\}} 1$ are equal to the number of walks of length 2 starting from nodes $v$ and $u$, respectively. And more generally, for $k = K$, $\sum_{w \in \mathcal{N}(v) \cup \{v\}} \sum_{x \in \mathcal{N}(w) \cup \{w\}} \cdots \sum_{z \in \mathcal{N}(y) \cup \{y\}} 1$ and $\sum_{w \in \mathcal{N}(u) \cup \{u\}} \sum_{x \in \mathcal{N}(w) \cup \{w\}} \cdots \sum_{z \in \mathcal{N}(y) \cup \{y\}} 1$ are equal to the number of walks of length $K$ starting from nodes $v$ and $u$, respectively. Thus, the representations learned by GIN-0 are related to the number of walks emanating from each node.

## B  PROOF OF THEOREM 4.2

We assume that the nodes of all graphs are annotated with $d$-dimensional feature vectors. No assumptions are made about those vectors. In what follows, we make use of the following inequality: $\|\sum_{i=1}^n \mathbf{x}_i\|_2 \leq \sum_{i=1}^n \|\mathbf{x}_i\|_2$ where $\mathbf{x}_i \in \mathbb{R}^d \ \forall i \in [n]$. We will also assume that we know the solution of the optimization problem of equation 1 and re-arrange the terms such that nodes that are matched with each other form pairs. For example, let $\mathcal{N}(v) = \{u_1, u_2, u_3, u_4, u_5\}$ and $\mathcal{N}(v') = \{u'_1, u'_2, u'_3, u'_4\}$. Suppose that nodes $u_1, u_3, u_4$ are macthed with nodes $u'_2, u'_4, u'_3$, respectively, while the rest of the nodes are left unmatched. Then, we would have that:

$$d = \left\| \sum_{u \in \mathcal{N}(v)} \mathbf{h}_u^{(k)} - \sum_{u' \in \mathcal{N}(v')} \mathbf{h}_{u'}^{(k)} \right\|_2$$

$$= \left\| \left( \mathbf{h}_{u_1}^{(k)} - \mathbf{h}_{u'_2}^{(k)} \right) + \left( \mathbf{h}_{u_3}^{(k)} - \mathbf{h}_{u'_4}^{(k)} \right) + \left( \mathbf{h}_{u_4}^{(k)} - \mathbf{h}_{u'_3}^{(k)} \right) + \mathbf{h}_{u_2}^{(k)} + \mathbf{h}_{u_5}^{(k)} - \mathbf{h}_{u'_1}^{(k)} \right\|_2$$

$$\leq \left\| \mathbf{h}_{u_1}^{(k)} - \mathbf{h}_{u'_2}^{(k)} \right\|_2 + \left\| \mathbf{h}_{u_3}^{(k)} - \mathbf{h}_{u'_4}^{(k)} \right\|_2 + \left\| \mathbf{h}_{u_4}^{(k)} - \mathbf{h}_{u'_3}^{(k)} \right\|_2 + \left\| \mathbf{h}_{u_2}^{(k)} \right\|_2 + \left\| \mathbf{h}_{u_5}^{(k)} \right\|_2 + \left\| \mathbf{h}_{u'_1}^{(k)} \right\|_2$$

$$(5)$$

### B.1  DGCNN

Let $L_f^{(k)}$ denote the Lipschitz constant associated with the fully connected layer of the $k$-th neighborhood aggregation layer of DGCNN. We set $d(v) = 1 + \deg(v)$ and also denote the set $\mathcal{N}(v) \cup \{v\}$ by $\tilde{\mathcal{N}}(v)$. We also choose the ReLU as the model's activation function. Then, we have:

$$\|\mathbf{h}_v^{(1)} - \mathbf{h}_{v'}^{(1)}\|_2 = \left\| \mathrm{ReLU} \left( \sum_{u \in \tilde{\mathcal{N}}(v)} \frac{\mathbf{W}^{(1)} \mathbf{h}_u^{(0)}}{d(v)} \right) - \mathrm{ReLU} \left( \sum_{u' \in \tilde{\mathcal{N}}(v')} \frac{\mathbf{W}^{(1)} \mathbf{h}_{u'}^{(0)}}{d(v')} \right) \right\|_2$$

$$\leq L_f^{(1)} \left\| \sum_{u \in \tilde{\mathcal{N}}(v)} \frac{\mathbf{h}_u^{(0)}}{d(v)} - \sum_{u' \in \tilde{\mathcal{N}}(v')} \frac{\mathbf{h}_{u'}^{'(0)}}{d(v')} \right\|_2$$

$$\leq L_f^{(1)} \text{WD}\Big( \bar{\mathcal{W}}_U^{(1)}(v), \bar{\mathcal{W}}_U^{(1)}(v') \Big)$$

$$||\mathbf{h}_v^{(2)} - \mathbf{h}_{v'}^{(2)}||_2 = \left\| \text{ReLU}\left( \sum_{u \in \tilde{\mathcal{N}}(v)} \frac{\mathbf{W}^{(2)} \mathbf{h}_u^{(1)}}{d(v)} \right) - \text{ReLU}\left( \sum_{u' \in \tilde{\mathcal{N}}(v')} \frac{\mathbf{W}^{(2)} \mathbf{h}_{u'}^{(1)}}{d(v')} \right) \right\|_2$$

$$= \left\| \text{ReLU}\left( \mathbf{W}^{(2)} \sum_{u \in \tilde{\mathcal{N}}(v)} \frac{\mathbf{h}_u^{(1)}}{d(v)} \right) - \text{ReLU}\left( \mathbf{W}^{(2)} \sum_{u' \in \tilde{\mathcal{N}}(v')} \frac{\mathbf{h}_{u'}^{(1)}}{d(v')} \right) \right\|_2$$

$$\leq L_f^{(2)} \left\| \sum_{u \in \tilde{\mathcal{N}}(v)} \frac{1}{d(v)} \text{ReLU}\left( \sum_{w \in \tilde{\mathcal{N}}(u)} \frac{\mathbf{W}^{(1)} \mathbf{h}_w^{(0)}}{d(u)} \right) - \sum_{u' \in \tilde{\mathcal{N}}(v')} \frac{1}{d(v')} \text{ReLU}\left( \sum_{w' \in \tilde{\mathcal{N}}(u')} \frac{\mathbf{W}^{(1)} \mathbf{h}_{w'}^{(0)}}{d(v')} \right) \right\|_2$$

$$= L_f^{(2)} \left\| \sum_{u \in \tilde{\mathcal{N}}(v)} \text{ReLU}\left( \mathbf{W}^{(1)} \sum_{w \in \tilde{\mathcal{N}}(u)} \frac{\mathbf{h}_w^{(0)}}{d(v)d(u)} \right) - \sum_{u' \in \tilde{\mathcal{N}}(v')} \text{ReLU}\left( \mathbf{W}^{(1)} \sum_{w' \in \tilde{\mathcal{N}}(u')} \frac{\mathbf{h}_{w'}^{(0)}}{d(v')d(u')} \right) \right\|_2$$

$$\leq L_f^{(2)} \left( \left\| \text{ReLU}\left( \mathbf{W}^{(1)} \sum_{w \in \tilde{\mathcal{N}}(u_1)} \frac{\mathbf{h}_w^{(0)}}{d(v)d(u_1)} \right) - \text{ReLU}\left( \mathbf{W}^{(1)} \sum_{w' \in \tilde{\mathcal{N}}(u'_1)} \frac{\mathbf{h}_{w'}^{(0)}}{d(v')d(u'_1)} \right) \right\|_2 + \dots \right.$$

$$\left. \dots + \left\| \text{ReLU}\left( \mathbf{W}^{(1)} \sum_{w \in \tilde{\mathcal{N}}(u_m)} \frac{\mathbf{h}_w^{(0)}}{d(v)d(u_m)} \right) - \text{ReLU}\left( \mathbf{W}^{(1)} \sum_{w' \in \tilde{\mathcal{N}}(u'_m)} \frac{\mathbf{h}_{w'}^{(0)}}{d(v')d(u'_m)} \right) \right\|_2 \right)$$

$$\leq L_f^{(2)} \left( L_f^{(1)} \left\| \sum_{w \in \tilde{\mathcal{N}}(u_1)} \frac{\mathbf{h}_w^{(0)}}{d(v)d(u_1)} - \sum_{w' \in \tilde{\mathcal{N}}(u'_1)} \frac{\mathbf{h}_{w'}^{(0)}}{d(v')d(u'_1)} \right\|_2 + \dots \right.$$

$$\left. \dots + L_f^{(1)} \left\| \sum_{w \in \tilde{\mathcal{N}}(u_m)} \frac{\mathbf{h}_w^{(0)}}{d(v)d(u_m)} - \sum_{w' \in \tilde{\mathcal{N}}(u'_m)} \frac{\mathbf{h}_{w'}^{(0)}}{d(v')d(u'_m)} \right\|_2 \right)$$

$$= L_f^{(2)} L_f^{(1)} \left( \left\| \sum_{w \in \tilde{\mathcal{N}}(u_1)} \frac{\mathbf{h}_w^{(0)}}{d(v)d(u_1)} - \sum_{w' \in \tilde{\mathcal{N}}(u'_1)} \frac{\mathbf{h}_{w'}^{(0)}}{d(v')d(u'_1)} \right\|_2 + \dots \right.$$

$$\left. \dots + \left\| \sum_{w \in \tilde{\mathcal{N}}(u_m)} \frac{\mathbf{h}_w^{(0)}}{d(v)d(u_m)} - \sum_{w' \in \tilde{\mathcal{N}}(u'_m)} \frac{\mathbf{h}_{w'}^{(0)}}{d(v')d(u'_m)} \right\|_2 \right)$$

$$= L_f^{(2)} L_f^{(1)} \text{WD}\Big( \bar{\mathcal{W}}_U^{(2)}(v), \bar{\mathcal{W}}_U^{(2)}(u) \Big)$$

$$\vdots$$

$$||\mathbf{h}_v^{(K)} - \mathbf{h}_{v'}^{(K)}||_2 = \left\| \text{ReLU}\left( \sum_{u \in \tilde{\mathcal{N}}(v)} \frac{\mathbf{W}^{(K)} \mathbf{h}_u^{(K-1)}}{d(v)} \right) - \text{ReLU}\left( \sum_{u' \in \tilde{\mathcal{N}}(v')} \frac{\mathbf{W}^{(K)} \mathbf{h}_{u'}^{(K-1)}}{d(v')} \right) \right\|_2$$

$$= \left\| \text{ReLU}\left( \mathbf{W}^{(K)} \sum_{u \in \tilde{\mathcal{N}}(v)} \frac{\mathbf{h}_u^{(K-1)}}{d(v)} \right) - \text{ReLU}\left( \mathbf{W}^{(K)} \sum_{u' \in \tilde{\mathcal{N}}(v')} \frac{\mathbf{h}_{u'}^{(K-1)}}{d(v')} \right) \right\|_2$$

$$\leq L_f^{(K)} \left\| \sum_{u \in \tilde{\mathcal{N}}(v)} \frac{1}{d(v)} \text{ReLU}\left( \sum_{w \in \tilde{\mathcal{N}}(u)} \frac{\mathbf{W}^{(K-1)} \mathbf{h}_w^{(K-2)}}{d(u)} \right) \right.$$

$$\left. - \sum_{u' \in \tilde{\mathcal{N}}(v')} \frac{1}{d(v')} \text{ReLU}\left( \sum_{w' \in \tilde{\mathcal{N}}(u')} \frac{\mathbf{W}^{(K-1)} \mathbf{h}_{w'}^{(K-2)}}{d(u')} \right) \right\|_2$$

$$= L_f^{(K)} \left\| \sum_{u \in \tilde{\mathcal{N}}(v)} \text{ReLU}\left( \mathbf{W}^{(K-1)} \sum_{w \in \tilde{\mathcal{N}}(u)} \frac{\mathbf{h}_w^{(K-2)}}{d(v)d(u)} \right) - \sum_{u' \in \tilde{\mathcal{N}}(v')} \text{ReLU}\left( \mathbf{W}^{(K-1)} \sum_{w' \in \tilde{\mathcal{N}}(u')} \frac{\mathbf{h}_{w'}^{(K-2)}}{d(v')d(u')} \right) \right\|_2$$

$$\leq L_f^{(K)} L_f^{(K-1)} \dots L_f^{(2)} \left( \left\| \frac{1}{d(v)d(u)\dots} \mathrm{ReLU}\left( \sum_{a\in\tilde{\mathcal{N}}(b)} \frac{\mathbf{W}^{(1)}\mathbf{h}_a^{(0)}}{d(b)} \right) \right. \right.$$

$$\left. \left. - \frac{1}{d(v')d(u')\dots} \mathrm{ReLU}\left( \sum_{a'\in\tilde{\mathcal{N}}(b')} \frac{\mathbf{W}^{(1)}\mathbf{h}_{a'}^{(0)}}{d(b')} \right) \right\|_2 + \dots \right)$$

$$= L_f^{(K)} L_f^{(K-1)} \dots L_f^{(2)} \left( \left\| \mathrm{ReLU}\left( \mathbf{W}^{(1)} \sum_{a\in\tilde{\mathcal{N}}(b)} \frac{\mathbf{h}_a^{(0)}}{d(v)d(u)\dots d(b)} \right) \right. \right.$$

$$\left. \left. - \mathrm{ReLU}\left( \mathbf{W}^{(1)} \sum_{a'\in\tilde{\mathcal{N}}(b')} \frac{\mathbf{h}_{a'}^{(0)}}{d(v')d(u')\dots d(b')} \right) \right\|_2 + \dots \right)$$

$$\leq L_f^{(K)} L_f^{(K-1)} \dots L_f^{(2)} \left( L_f^{(1)} \left\| \sum_{a\in\tilde{\mathcal{N}}(b)} \frac{\mathbf{h}_a^{(0)}}{d(v)d(u)\dots d(b)} - \sum_{a'\in\tilde{\mathcal{N}}(b')} \frac{\mathbf{h}_{a'}^{(0)}}{d(v')d(u')\dots d(b')} \right\|_2 + \dots \right)$$

$$= \leq L_f^{(K)} L_f^{(K-1)} \dots L_f^{(1)} \left( \left\| \sum_{a\in\tilde{\mathcal{N}}(b)} \frac{\mathbf{h}_a^{(0)}}{d(v)d(u)\dots d(b)} - \sum_{a'\in\tilde{\mathcal{N}}(b')} \frac{\mathbf{h}_{a'}^{(0)}}{d(v')d(u')\dots d(b')} \right\|_2 + \dots \right)$$

$$= L_f^{(K)} L_f^{(K-1)} \dots L_f^{(1)} \mathrm{WD}\left( \bar{\mathcal{W}}^{(K)}(v), \bar{\mathcal{W}}^{(K)}(v') \right)$$

Note that in the above formulation, we have assumed that the problem of equation 1 is minimized when nodes $v, u, b$ and $a$ are matched with nodes $v', u', b'$ and $a'$, respectively, while for clarity of presentation, the rest of the matched nodes and the unmatched nodes are omitted.

## B.2 GAT

In the previous section, we assumed that DGCNN is applied to an unweighted graph and that the random walk is uniform, i.e., every neighbor of the current node has an equal chance of being selected as the next node in the walk. The GAT model instead employs an attention mechanism to assign weights to the edges of the graph. The weight of the edge from node $v$ to node $u$ is denoted by $\alpha_{vu}$ and we have that $\sum_{u\in\mathcal{N}(v)} \alpha_{vu} = 1$. Therefore, the analysis of the previous section also holds for GAT. Instead of weighting the vector $\mathbf{h}_\mathrm{w}$ of a walk w by the product of the inverse of the degrees of the nodes along the walk, the vector is weighted by the product of the learned attention coefficients.

## B.3 GCN

Let $L_f^{(k)}$ denote the Lipschitz constant associated with the fully connected layer of the $k$-th neighborhood aggregation layer of GCN. We set $d(v) = 1 + \deg(v)$ and also denote the set $\mathcal{N}(v) \cup \{v\}$ by $\tilde{\mathcal{N}}(v)$. Then, we have:

$$\|\mathbf{h}_v^{(1)} - \mathbf{h}_{v'}^{(1)}\|_2 = \left\| \mathrm{ReLU}\left( \sum_{u\in\tilde{\mathcal{N}}(v)} \frac{\mathbf{W}^{(1)}\mathbf{h}_u^{(0)}}{\sqrt{d(v)d(u)}} \right) - \mathrm{ReLU}\left( \sum_{u'\in\tilde{\mathcal{N}}(v')} \frac{\mathbf{W}^{(1)}\mathbf{h}_{u'}^{(0)}}{\sqrt{d(v')d(u')}} \right) \right\|_2$$

$$\leq L_f^{(1)} \left\| \sum_{u\in\tilde{\mathcal{N}}(v)} \frac{\mathbf{h}_u^{(0)}}{\sqrt{d(v)d(u)}} - \sum_{u'\in\tilde{\mathcal{N}}(v')} \frac{\mathbf{h}_{u'}^{(0)}}{\sqrt{d(v')d(u')}} \right\|_2$$

$$\leq L_f^{(1)} \mathrm{WD}\left( \tilde{\mathcal{W}}_U^{(1)}(v), \tilde{\mathcal{W}}_U^{(1)}(v') \right)$$

$$\|\mathbf{h}_v^{(2)} - \mathbf{h}_{v'}^{(2)}\|_2 = \left\| \mathrm{ReLU}\left( \sum_{u\in\tilde{\mathcal{N}}(v)} \frac{\mathbf{W}^{(2)}\mathbf{h}_u^{(1)}}{\sqrt{d(v')d(u')}} \right) - \mathrm{ReLU}\left( \sum_{u'\in\tilde{\mathcal{N}}(v')} \frac{\mathbf{W}^{(2)}\mathbf{h}_{u'}^{(1)}}{\sqrt{d(v')d(u')}} \right) \right\|_2$$

$$= \left\| \text{ReLU}\left( \mathbf{W}^{(2)} \sum_{u \in \tilde{\mathcal{N}}(v)} \frac{\mathbf{h}_u^{(1)}}{\sqrt{d(v)d(u)}} \right) - \text{ReLU}\left( \mathbf{W}^{(2)} \sum_{u' \in \tilde{\mathcal{N}}(v')} \frac{\mathbf{h}_{u'}^{(1)}}{\sqrt{d(v')d(u')}} \right) \right\|_2$$

$$\leq L_f^{(2)} \left\| \sum_{u \in \tilde{\mathcal{N}}(v)} \frac{1}{\sqrt{d(v)d(u)}} \text{ReLU}\left( \sum_{w \in \tilde{\mathcal{N}}(u)} \frac{\mathbf{W}^{(1)} \mathbf{h}_w^{(0)}}{\sqrt{d(u)d(w)}} \right) \right.$$

$$\left. - \sum_{u' \in \tilde{\mathcal{N}}(v')} \frac{1}{\sqrt{d(v)d(u)}} \text{ReLU}\left( \sum_{w' \in \tilde{\mathcal{N}}(u')} \frac{\mathbf{W}^{(1)} \mathbf{h}_{w'}^{(0)}}{\sqrt{d(u')d(w')}} \right) \right\|_2$$

$$= L_f^{(2)} \left\| \sum_{u \in \tilde{\mathcal{N}}(v)} \text{ReLU}\left( \mathbf{W}^{(1)} \sum_{w \in \tilde{\mathcal{N}}(u)} \frac{\mathbf{h}_w^{(0)}}{d(u)\sqrt{d(v)d(w)}} \right) \right.$$

$$\left. - \sum_{u' \in \tilde{\mathcal{N}}(v')} \text{ReLU}\left( \mathbf{W}^{(1)} \sum_{w' \in \tilde{\mathcal{N}}(u')} \frac{\mathbf{h}_{w'}^{(0)}}{d(u')\sqrt{d(v')d(w')}} \right) \right\|_2$$

$$\leq L_f^{(2)} \left( \left\| \sigma\left( \mathbf{W}^{(1)} \sum_{w \in \tilde{\mathcal{N}}(u_1)} \frac{\mathbf{h}_w^{(0)}}{d(u_1)\sqrt{d(v)d(w)}} \right) - \sigma\left( \mathbf{W}^{(1)} \sum_{w' \in \tilde{\mathcal{N}}(u_1')} \frac{\mathbf{h}_{w'}^{(0)}}{d(u_1')\sqrt{d(v')d(w')}} \right) \right\|_2 + \dots \right.$$

$$\left. \dots + \left\| \sigma\left( \mathbf{W}^{(1)} \sum_{w \in \tilde{\mathcal{N}}(u_m)} \frac{\mathbf{h}_w^{(0)}}{d(u_m)\sqrt{d(v)d(w)}} \right) - \sigma\left( \mathbf{W}^{(1)} \sum_{w' \in \tilde{\mathcal{N}}(u_m')} \frac{\mathbf{h}_{w'}^{(0)}}{d(u_m')\sqrt{d(v')d(w')}} \right) \right\|_2 \right)$$

$$\leq L_f^{(2)} \left( L_f^{(1)} \left\| \sum_{w \in \tilde{\mathcal{N}}(u_1)} \frac{\mathbf{h}_w^{(0)}}{d(u_1)\sqrt{d(v)d(w)}} - \sum_{w' \in \tilde{\mathcal{N}}(u_1')} \frac{\mathbf{h}_{w'}^{(0)}}{d(u_1')\sqrt{d(v')d(w')}} \right\|_2 + \dots \right.$$

$$\left. \dots + L_f^{(1)} \left\| \sum_{w \in \tilde{\mathcal{N}}(u_m)} \frac{\mathbf{h}_w^{(0)}}{d(u_m)\sqrt{d(v)d(w)}} - \sum_{w' \in \tilde{\mathcal{N}}(u_m')} \frac{\mathbf{h}_{w'}^{(0)}}{d(u_m')\sqrt{d(v')d(w')}} \right\|_2 \right)$$

$$= L_f^{(2)} L_f^{(1)} \left( \left\| \sum_{w \in \tilde{\mathcal{N}}(u_1)} \frac{\mathbf{h}_w^{(0)}}{d(u_1)\sqrt{d(v)d(w)}} - \sum_{w' \in \tilde{\mathcal{N}}(u_1')} \frac{\mathbf{h}_{w'}^{(0)}}{d(u_1')\sqrt{d(v')d(w')}} \right\|_2 + \dots \right.$$

$$\left. \dots + \left\| \sum_{w \in \tilde{\mathcal{N}}(u_m)} \frac{\mathbf{h}_w^{(0)}}{d(u_m)\sqrt{d(v)d(w)}} - \sum_{w' \in \tilde{\mathcal{N}}(u_m')} \frac{\mathbf{h}_{w'}^{(0)}}{d(u_m')\sqrt{d(v')d(w')}} \right\|_2 \right)$$

$$= L_f^{(2)} L_f^{(1)} \text{WD}\left( \tilde{\mathcal{W}}^{(2)}(v), \tilde{\mathcal{W}}^{(2)}(u) \right)$$

$$\vdots$$

$$\|\mathbf{h}_v^{(K)} - \mathbf{h}_{v'}^{(K)}\|_2 = \left\| \text{ReLU}\left( \sum_{u \in \tilde{\mathcal{N}}(v)} \frac{\mathbf{W}^{(K)} \mathbf{h}_u^{(K-1)}}{\sqrt{d(v)d(u)}} \right) - \text{ReLU}\left( \sum_{u' \in \tilde{\mathcal{N}}(v')} \frac{\mathbf{W}^{(K)} \mathbf{h}_{u'}^{(K-1)}}{\sqrt{d(v')d(u')}} \right) \right\|_2$$

$$= \left\| \text{ReLU}\left( \mathbf{W}^{(K)} \sum_{u \in \tilde{\mathcal{N}}(v)} \frac{\mathbf{h}_u^{(K-1)}}{\sqrt{d(v)d(u)}} \right) - \text{ReLU}\left( \mathbf{W}^{(K)} \sum_{u' \in \tilde{\mathcal{N}}(v')} \frac{\mathbf{h}_{u'}^{(K-1)}}{\sqrt{d(v')d(u')}} \right) \right\|_2$$

$$\leq L_f^{(K)} \left\| \sum_{u \in \tilde{\mathcal{N}}(v)} \frac{1}{\sqrt{d(v)d(u)}} \text{ReLU}\left( \sum_{w \in \tilde{\mathcal{N}}(u)} \frac{\mathbf{W}^{(K-1)} \mathbf{h}_w^{(K-2)}}{\sqrt{d(u)d(w)}} \right) \right.$$

$$\left. - \sum_{u' \in \tilde{\mathcal{N}}(v')} \frac{1}{\sqrt{d(v')d(u')}} \text{ReLU}\left( \sum_{w' \in \tilde{\mathcal{N}}(u')} \frac{\mathbf{W}^{(K-1)} \mathbf{h}_{w'}^{(K-2)}}{\sqrt{d(u')d(w')}} \right) \right\|_2$$

$$= L_f^{(K)} \left\| \sum_{u \in \tilde{\mathcal{N}}(v)} \text{ReLU}\left( \mathbf{W}^{(K-1)} \sum_{w \in \tilde{\mathcal{N}}(u)} \frac{\mathbf{h}_w^{(K-2)}}{d(u)\sqrt{d(v)d(w)}} \right) \right.$$

$$- \sum_{u' \in \tilde{\mathcal{N}}(v')} \text{ReLU}\left(\mathbf{W}^{(K-1)} \sum_{w' \in \tilde{\mathcal{N}}(u')} \frac{\mathbf{h}_{w'}^{(K-2)}}{d(u')\sqrt{d(v')d(w')}}\right)\Bigg|\Bigg|_2$$

$$\leq L_f^{(K)} L_f^{(K-1)} \dots L_f^{(2)} \Bigg(\Bigg|\Bigg| \frac{1}{d(u)\dots\sqrt{d(v)d(b)}} \text{ReLU}\left(\sum_{a \in \tilde{\mathcal{N}}(b)} \frac{\mathbf{W}^{(1)}\mathbf{h}_a^{(0)}}{\sqrt{d(b)d(a)}}\right)$$

$$- \frac{1}{d(u')\dots\sqrt{d(v')d(b')}} \text{ReLU}\left(\sum_{a' \in \tilde{\mathcal{N}}(b')} \frac{\mathbf{W}^{(1)}\mathbf{h}_{a'}^{(0)}}{\sqrt{d(b')d(a')}}\right)\Bigg|\Bigg|_2 + \dots\Bigg)$$

$$= L_f^{(K)} L_f^{(K-1)} \dots L_f^{(2)} \Bigg(\Bigg|\Bigg| \text{ReLU}\left(\mathbf{W}^{(1)} \sum_{a \in \tilde{\mathcal{N}}(b)} \frac{\mathbf{h}_a^{(0)}}{d(u)\dots d(b)\sqrt{d(v)d(a)}}\right)$$

$$- \text{ReLU}\left(\mathbf{W}^{(1)} \sum_{a' \in \tilde{\mathcal{N}}(b')} \frac{\mathbf{h}_{a'}^{(0)}}{d(u')\dots d(b')\sqrt{d(v')d(a')}}\right)\Bigg|\Bigg|_2 + \dots\Bigg)$$

$$\leq L_f^{(K)} L_f^{(K-1)} \dots L_f^{(2)} \Bigg(L_f^{(1)}\Bigg|\Bigg| \sum_{a \in \tilde{\mathcal{N}}(b)} \frac{\mathbf{h}_a^{(0)}}{d(u)\dots d(b)\sqrt{d(v)d(a)}}$$

$$- \sum_{a' \in \tilde{\mathcal{N}}(b')} \frac{\mathbf{h}_{a'}^{(0)}}{d(u')\dots d(b')\sqrt{d(v')d(a')}}\Bigg|\Bigg|_2 + \dots\Bigg)$$

$$= \leq L_f^{(K)} L_f^{(K-1)} \dots L_f^{(1)} \Bigg(\Bigg|\Bigg| \sum_{a \in \tilde{\mathcal{N}}(b)} \frac{\mathbf{h}_a^{(0)}}{d(u)\dots d(b)\sqrt{d(v)d(a)}}$$

$$- \sum_{a' \in \tilde{\mathcal{N}}(b')} \frac{\mathbf{h}_{a'}^{(0)}}{d(u')\dots d(b')\sqrt{d(v')d(a')}}\Bigg|\Bigg|_2 + \dots\Bigg)$$

$$= L_f^{(K)} L_f^{(K-1)} \dots L_f^{(1)} \text{WD}\left(\tilde{\mathcal{W}}^{(K)}(v), \tilde{\mathcal{W}}^{(K)}(v')\right)$$

### B.4 GIN-0

The GIN-0 model updates node representations as shown in Equation equation 3. We also make once again Assumption equation A.1.

Let $L_f^{(k)}$ denote the Lipschitz constant associated with the MLP of the $k$-th neighborhood aggregation layer of GIN-0. We denote the set $\mathcal{N}(v) \cup \{v\}$ by $\tilde{\mathcal{N}}(v)$. Then, we have:

$$||\mathbf{h}_v^{(1)} - \mathbf{h}_{v'}^{(1)}||_2 = \Bigg|\Bigg| \text{MLP}^{(1)}\left(\sum_{u \in \tilde{\mathcal{N}}(v)} \mathbf{h}_u^{(0)}\right) - \text{MLP}^{(1)}\left(\sum_{u' \in \tilde{\mathcal{N}}(v')} \mathbf{h}_{u'}^{(0)}\right)\Bigg|\Bigg|_2$$

$$\leq L_f^{(1)}\Bigg|\Bigg| \sum_{u \in \tilde{\mathcal{N}}(v)} \mathbf{h}_u^{(0)} - \sum_{u' \in \tilde{\mathcal{N}}(v')} \mathbf{h}_{u'}^{(0)}\Bigg|\Bigg|_2$$

$$\leq L_f^{(1)}\Bigg|\Bigg| \text{WD}\left(\mathcal{W}^{(1)}(v), \mathcal{W}^{(1)}(v')\right)\Bigg|\Bigg|_2$$

$$||\mathbf{h}_v^{(2)} - \mathbf{h}_{v'}^{(2)}||_2 = \Bigg|\Bigg| \text{MLP}^{(2)}\left(\sum_{w \in \tilde{\mathcal{N}}(v)} \mathbf{h}_w^{(1)}\right) - \text{MLP}^{(2)}\left(\sum_{w \in \tilde{\mathcal{N}}(u)} \mathbf{h}_w^{(1)}\right)\Bigg|\Bigg|_2$$

$$\leq L_f^{(2)} \left\| \sum_{u \in \tilde{\mathcal{N}}(v)} \text{MLP}^{(1)}\left( \sum_{w \in \tilde{\mathcal{N}}(u)} \mathbf{h}_w^{(0)} \right) - \sum_{u' \in \tilde{\mathcal{N}}(v')} \text{MLP}^{(1)}\left( \sum_{w' \in \tilde{\mathcal{N}}(u')} \mathbf{h}_{w'}^{(0)} \right) \right\|_2$$

$$\leq L_f^{(2)} \left( \left\| \text{MLP}^{(1)}\left( \sum_{w \in \tilde{\mathcal{N}}(u_1)} \mathbf{h}_w^{(0)} \right) - \text{MLP}^{(1)}\left( \sum_{w' \in \tilde{\mathcal{N}}(u_1')} \mathbf{h}_{w'}^{(0)} \right) \right\|_2 + \ldots \right.$$

$$\left. + \left\| \text{MLP}^{(1)}\left( \sum_{w \in \tilde{\mathcal{N}}(u_m)} \mathbf{h}_w^{(0)} \right) - \text{MLP}^{(1)}\left( \sum_{w' \in \tilde{\mathcal{N}}(u_m')} \mathbf{h}_{w'}^{(0)} \right) \right\|_2 \right)$$

$$\leq L_f^{(2)} \left( L_f^{(1)} \left\| \sum_{w \in \tilde{\mathcal{N}}(u_1)} \mathbf{h}_w^{(0)} - \sum_{w' \in \tilde{\mathcal{N}}(u_1')} \mathbf{h}_{w'}^{(0)} \right\|_2 + \ldots + L_f^{(1)} \left\| \sum_{w \in \tilde{\mathcal{N}}(u_m)} \mathbf{h}_x^{(0)} - \sum_{w' \in \tilde{\mathcal{N}}(u_m')} \mathbf{h}_{w'}^{(0)} \right\|_2 \right)$$

$$= L_f^{(2)} L_f^{(1)} \left( \left\| \sum_{w \in \tilde{\mathcal{N}}(u_1)} \mathbf{h}_w^{(0)} - \sum_{w' \in \tilde{\mathcal{N}}(u_1')} \mathbf{h}_{w'}^{(0)} \right\|_2 + \ldots + \left\| \sum_{w \in \tilde{\mathcal{N}}(u_m)} \mathbf{h}_w^{(0)} - \sum_{w' \in \tilde{\mathcal{N}}(u_m')} \mathbf{h}_{w'}^{(0)} \right\|_2 \right)$$

$$\leq L_f^{(2)} L_f^{(1)} \text{WD}\left( \mathcal{W}^{(2)}(v), \mathcal{W}^{(2)}(v') \right)$$

$$\vdots$$

$$\|\mathbf{h}_v^{(K)} - \mathbf{h}_{v'}^{(K)}\|_2 = \left\| \text{MLP}^{(K)}\left( \sum_{u \in \tilde{\mathcal{N}}(v)} \mathbf{h}_u^{(K-1)} \right) - \text{MLP}^{(K)}\left( \sum_{u' \in \tilde{\mathcal{N}}(v')} \mathbf{h}_{u'}^{(K-1)} \right) \right\|_2$$

$$\leq L_f^{(K)} \left\| \sum_{u \in \tilde{\mathcal{N}}(v)} \text{MLP}^{(K-1)}\left( \sum_{w \in \tilde{\mathcal{N}}(u)} \mathbf{h}_w^{(K-2)} \right) - \sum_{u' \in \tilde{\mathcal{N}}(v')} \text{MLP}^{(K-1)}\left( \sum_{w' \in \tilde{\mathcal{N}}(u')} \mathbf{h}_{w'}^{(K-2)} \right) \right\|_2$$

$$\leq L_f^{(K)} L_f^{(K-1)} \ldots L_f^{(2)} \left( \left\| \text{MLP}^{(1)}\left( \sum_{a \in \tilde{\mathcal{N}}(b)} \mathbf{h}_a^{(0)} \right) - \text{MLP}^{(1)}\left( \sum_{a' \in \tilde{\mathcal{N}}(b')} \mathbf{h}_{a'}^{(0)} \right) \right\|_2 + \ldots \right)$$

$$\leq L_f^{(K)} L_f^{(K-1)} \ldots L_f^{(2)} \left( L_f^{(1)} \left\| \sum_{a \in \tilde{\mathcal{N}}(b)} \mathbf{h}_a^{(0)} - \sum_{a' \in \tilde{\mathcal{N}}(b')} \mathbf{h}_{a'}^{(0)} \right\|_2 + \ldots \right)$$

$$= \leq L_f^{(K)} L_f^{(K-1)} \ldots L_f^{(1)} \left( \left\| \sum_{a \in \tilde{\mathcal{N}}(b)} \mathbf{h}_a^{(0)} - \sum_{a' \in \tilde{\mathcal{N}}(b')} \mathbf{h}_{a'}^{(0)} \right\|_2 + \ldots \right)$$

$$= L_f^{(K)} L_f^{(K-1)} \ldots L_f^{(1)} \text{WD}\left( \mathcal{W}^{(K)}(v), \mathcal{W}^{(K)}(v') \right)$$

