# OpenReview forum: "What Do GNNs Actually Learn? Towards Understanding their Representations"
_ICLR.cc/2024/Conference — ICLR 2024 Conference Withdrawn Submission_

### Official Review · Reviewer_c5EH · 2023-10-24

**Soundness:** 2 fair
**Presentation:** 3 good
**Contribution:** 2 fair
**Rating:** 3
**Confidence:** 3

**Summary:**

This paper studies what graph structural properties are encoded in node representations by GNNs. Specifically, the authors consider two cases where node features are defined as a constant and general feature vectors: in the former case, the distance of representations of two nodes are bounded by the difference of number of walks starting from those nodes; in the latter case, the bound is related with the walk distance between two nodes and connected with oversquashing.

**Strengths:**

1. The writing is good. Notations and assumptions are presented in a clear manner. It is also easy to follow the idea, and the results also align with our intuitions.
2. Theoretical results are consistent with empirical observations, though more details might need to be provided.
3. The link to over-squashing from the perspective of walk distance seems interesting, though the conclusion may have already existed in previous works.

**Weaknesses:**

1. Analysis in 4.1 adopts a strong and unrealistic assumption that all nodes share the same feature. I do not fully understand why this setting is of theoretical interest or is relevant to practice, as it is perhaps obvious that GNNs will not perform good if nodes are identical and there exist trivial solutions such as adding one hot encodings. The conclusion about DGCNN and GAT might also have been overgeneralized, since this is not the case for other node features.
2. The bounds for both theorems do not seem tight and deteriorate with more layers. The proof circumvents nonlinearities and weights by simply resorting to an unknown Lipschitz constant for each layer, which could be arbitrarily bad in the worst case. PS, it is perhaps imprecise to term those representations as “learned representations” since no learning procedure is involved in the analysis (and thus the title “WHAT DO GNNS ACTUALLY LEARN” is also ambiguous.)
3. It is unclear what unexplained phenomenon can be explained by your analysis or how the analysis may help practice. While one could it a motivation to develop new models for improving representations of GNNs (just like those works in WL tests did), regrettably it is not the focus of this paper. Stronger theoretical results have already been established in research for oversmoothing (e.g. see [1] that is 4 year ago and many other follow-up works); while I understand the setting might be different and more variants are considered, results can still be adapted from those works.
4. For experiments, the phenomenon in figure 1 is based on restricted settings while the phenomenon in figure 2 is only reported on a single synthetic dataset. Additionally, the theorem seems infeasible to explain the strong correlation between node representation distances and number of walks reported in figure 1, as there is only upper bound but no lower bound given in the theorem.

Minors:
In theorem 4.2, h \mathbb R -> h \in \mathbb R
you might want to use $|\cdot|$ for scalar rather than l2 norm

[1] Graph Neural Networks Exponentially Lose Expressive Power for Node Classification, in ICLR 2020

**Questions:**

1. In experiments reported in figure 1, how are node features defined in those datasets?
2. Does the reported phenomenon in figure 1 and 3 holds for GNNs throughout the training process?
3. Can the proof in 4.1 extend to neural networks with bias terms, i.e. when MLP is not a homogeneous function as is usually the case?

---

### Official Review · Reviewer_dPEn · 2023-10-30

**Soundness:** 2 fair
**Presentation:** 2 fair
**Contribution:** 1 poor
**Rating:** 3
**Confidence:** 3

**Summary:**

This paper aims to study what kind of structural information is encoded into the node representations learned by four GNN models, including DGCNN, GAT, GCN, and GIN. Under the case of an input graph with identical input node features, the paper claims that k-layer GAT and DGCNN capture no structural properties of the neighborhoods of nodes and the representations that emerge at the k-layer of GCN and GIN are related to the number of normalized walks of length k over the graph. For an input graph with general node features, the paper claims that the representations that emerge at the k-layer of these four GNN models depend only on the nodes that can be reached in exactly k steps and not on the intermediate nodes along each walk.

**Strengths:**

1. The paper is clear and easy to follow.

2. The problem studied in the paper is important within the GNN community.

**Weaknesses:**

1. The novelty and the contributions of the paper are quite limited compared to existing work on the representation power of GNNs. Prior research, as indicated in references [1, 5, 6], has delved deeply into discerning the structural properties that GNNs can efficiently capture or fail to address. The idea of studying the representation of GNNs from the view of random walks or paths on graphs is also not new. [1] has studied the representation power of GCN through the looking glass of graph moments, a key property of graph topology encoding paths of various lengths. [6] has proved several important graph properties that rely entirely on local information, such as longest or shortest cycle, diameter, or clique information, cannot be computed by GNNs.

 2. Some important related works that also studied the representation powers of GNNs are missing, particularly [1], [2], [3], [4], [5], and [6], which raises concerns regarding the comprehensiveness and thoroughness of the literature review.

 3. The claim that DGCNN and GAT do not capture structural properties of the neighborhood of nodes might be NOT true. The proof A.1 and A.2 simply ignore the Training Dynamics of DGCNN and GAT. Given a subset of labeled nodes which are used for training GNN models, the error will backpropagate differently depending on the loss at these labeled nodes. This can lead to different learned representations for nodes close to the labeled ones versus those farther away, even if all initial features are identical. Therefore, this claim might not be true.

 4. The assumptions of the Lipschitz constants associated with the fully connected layer of the k-th neighborhood aggregation layer of GNN or MLP are too strong. Since this assumption is not a general Lipschitz assumption in GNN literature, it is not clear if such Lipchitz constants exist for all GNN layers. There is also no discussion on the validation of their Lipschitz assumptions.

 5. The assumed Lipschitz constants may inherently embed some graph information since it is associated with the aggregation layer. This association suggests the possibility of the Lipschitz constants encapsulating information from intermediate nodes along a walk. Therefore, the claims following Theorem 4.2 that the representations of all models learned at the k-th neighborhood aggregation layer depend only on the nodes that can be reached in exactly k steps and not on the intermediate nodes along each walk might NOT be true.

 6. The tightness of the Lipschitz bounds is unknown, which presents a significant concern. If these Lipschitz constants are substantially large, the cumulative effect from the product of all the constants across the k layers can lead to bounds that are exceedingly broad. Such expansive bounds might render them trivial or even ineffective in constraining the behavior of the GNNs as intended. Therefore, the theoretical results of the paper might not hold.

 7. The paper only focuses on DGCNN, GAT, GCN, and GIN, not sure if the results can be generalized to other types of GNNs, such as massage pass neural networks, APPNP, and GPRGNN.

Overall, the novelty and the contributions of the paper are quite limited compared to existing work on the representation power of GNNs, where existing works have shown some graph structural properties that GNNs can or cannot learn. Some Claims of the paper might not be true due to the proof either ignoring the training dynamics of GNNs or making strong Lipschitz assumptions. Given these concerns, the reviewer recommends rejection.


Reference

[1] Understanding the Representation Power of Graph Neural Networks in Learning Graph Topology, NeurIPS 2019.

[2] What Can Neural Networks Reason About? ICLR 2020.

[3] Graph Neural Networks Exponentially Loss Expressive Power for Node Representation, ICLR 2020.

[4] How hard is to distinguish graphs with graph neural networks? NeurIPS 2020.

[5] Can Graph Neural Networks Count Substructures? NeurIPS 2020.

[6] Generalization and Representational Limits of Graph Neural Networks, NeurIPS 2020.

**Questions:**

Please refer to the Weaknesses.

---

### Official Review · Reviewer_A5xe · 2023-11-01

**Soundness:** 3 good
**Presentation:** 2 fair
**Contribution:** 2 fair
**Rating:** 5
**Confidence:** 3

**Summary:**

This paper studies the node representations learned by GCN, DGCNN, GAT, and GIN-0. An interesting finding is that the distance between representations of some models is controlled by walks. The authors consider the cases when nodes are annotated with the same features or arbitrary features. Experiments support the theoretical findings.

**Strengths:**

1. As far as I know, the analysis is unique and interesting.
2. The theoretical proof seems correct.
3. The new understanding of over-squashing in terms of the graph topology is interesting.

**Weaknesses:**

1. The clarity is not so good for me, especially in Section 4.2. Please refer to "Questions" for details.
2. I am not so sure about the practical significance of this work. This work provides some new insights and understanding. It is better to generate new techniques from the insights to improve the model or mitigate the over-squashing.

**Questions:**

1. The definition of walk distance in Eqn.1 is complicated to me as an optimization problem. What does WD imply in a high-level understanding?

2. In Theorem 4.2, the parameters in WD in the three cases are also difficult to interpret. What does the three WD in Section 4.2 mean? Some simple examples can help the illustration.

3. There is a conclusion that "We further show that nodes with divergent structural characteristics can have similar representations when they share similar k-length walk patterns." Does it refer to a stricter version of over-smoothing since here it requires a similar k-length walk pattern? Please clarify.

---

### Official Review · Reviewer_K44H · 2023-11-02

**Soundness:** 2 fair
**Presentation:** 3 good
**Contribution:** 2 fair
**Rating:** 3
**Confidence:** 3

**Summary:**

This paper attempts to answer the following question: What types of (structural) knowledge are embedded into the final node features at the output of a graph neural network (GNN)? The authors first consider the case where all nodes have identical input features, in order to focus only on the graph structure (i.e., connectivity), and then proceed to the case where nodes have arbitrary features. Four different types of message-passing GNNs (GCN, GIN, GAT, and DGCNN) are analyzed, and the properties of their output node features are connected to random walks in the graph and Lipschitz constants of the intermediate aggregation functions. The analysis is also extended to shed light on GNNs' oversquashing phenomenon.

**Strengths:**

While GNNs have shown great empirical promise in learning on graph-structured data, their theoretical properties are not very well understood. The paper, therefore, addresses an important problem in identifying the learning properties of GNNs and comparison between different message-passing architectures.

**Weaknesses:**

I believe that the analysis is quite limited in terms of the overall contribution to the theoretical understanding of GNNs. Since the analysis only focuses on four of the numerous GNN architectures proposed in the literature over the past few years, it is unclear if and how it extends to other types of message-passing operations. Even for the GIN architecture that is among the ones analyzed in this work, there are several assumptions, such as $\epsilon = 0$ and the absence of the bias (it is unclear why the inequality $\\|\mathbf{b}\\| \ll \\|\mathbf{W}\mathbf{x}\\|$, mentioned after Assumption A.1, should hold in practice for any given $\mathbf{x}$). Also, the statement that "the representations learned at the $k$-th layer of the models are related to the initial features of nodes that can be reached in exactly $k$ steps." is trivial following the very definition of MPNNs on page 3, and should not be claimed as a contribution of the paper.

**Questions:**

- In Section 3.1, it would help to emphasize that each node can be visited any number of times on a given walk.

- Following the above comment, I wonder if your analysis can be extended to the case where a node's message does not get back to itself, such as in the PathNN architecture in the following work [A]:

  *[A] Michel, Gaspard, Giannis Nikolentzos, Johannes F. Lutzeyer, and Michalis Vazirgiannis. "Path neural networks: Expressive and accurate graph neural networks." In International Conference on Machine Learning, pp. 24737-24755. PMLR, 2023.*

- It seems that the results in Theorem 4.1 directly stem from the fact that the initial node features are equal. The **absence** of node features does not equate the **presence of identical** node features. How does replacing the identical features with **random** features (similar to the work [B] below) change the results?

  *[B] Kanatsoulis, Charilaos I., and Alejandro Ribeiro. "Representation Power of Graph Neural Networks: Improved Expressivity via Algebraic Analysis." arXiv preprint arXiv:2205.09801 (2022).*

- Could you comment on the tightness of the upper bounds in Theorems 4.1 and 4.2, and whether you can characterize any **lower** bounds on the distance of final representations?